# BOTTLENECK-GUIDED SPECTRAL SUBGOALS FOR OFFLINE GOAL-CONDITIONED RL

## ABSTRACT

Offline goal-conditioned RL (OGCRL) learns to reach arbitrary goals from offline dataset, but long-horizon performance hinges on crossing a handful of hard-to-cross bottlenecks. These bottlenecks not only dictate the feasible paths toward the goal but also act as critical keypoints, marking the transitions between adjacent regions and providing the agent with essential directional guidance. Prior hierarchical methods pick subgoals by time or short-horizon value heuristics, which do not localize the bottleneck, as a result, the agent losing the clear guidance that bottlenecks could provide about where to pass next. We instead model long-horizon planning as "cross the next bottleneck": we apply Laplacian spectral clustering to offline dataset to expose bottlenecks and then identify trajectories from the offline dataset that cross these boundaries, and the intersects are defined as keypoints (KPs). Then the most representative KPs are automatically selected and a directed KP reachability graph $\mathcal{G}_{\mathrm{KP}}$ is constructed based on the selected KPs. We then restrict high-level choices to these bottleneck states and use a pluggable low-level controller to execute the short transitions between them. We provide theory showing that under a standard metastable decomposition of the state space, routing through bottlenecks yields an (approximately) optimal one-step subgoal in terms of hitting-time, and that Laplacian spectra recover bottlenecks with high overlap. Thus, Laplacian spectral clustering can discover approximately optimal subgoals. Empirically, the same pattern holds: across D4RL and OGBench, our method achieves state-of-the-art results on a broad set of navigation and manipulation tasks and across diverse dataset regimes, for example, **96.5%** on **AntMaze** and **84.5%** on **Franka-Kitchen**.

## 1 INTRODUCTION

Long-horizon sparse rewards remain a core challenge for offline goal-conditioned reinforcement learning (OGCRL): datasets are limited and biased, online interaction is unavailable, and credit assignment couples with planning over long time scales. In most OGCRL tasks, the state space decomposes into well-connected regions (e.g., rooms and corridors in navigation) linked by a few hard-to-cross bottlenecks (e.g., doorways or narrow chokepoints in mazes). These bottlenecks act as structural keypoints that any successful trajectory must pass and thus provide

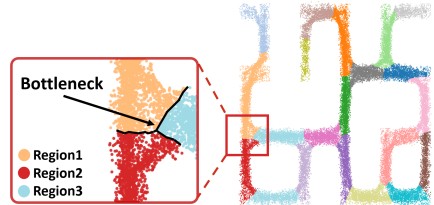

Figure 1: Laplacian spectral clustering in `pointmaze-large`. Colors indicate regions, boundaries align with hard-to-cross bottlenecks.

a clear high-level guidance signal: cross the next bottleneck to move from one region to the next. Fig. 1 illustrates our method's identification of regions and bottlenecks: each colored patch corresponds to a region, and the boundaries between patches align with bottlenecks.

Existing OGCRL approaches typically adopt *hierarchical* frameworks: a high level proposes a subgoal and a low level executes to reach it (Chane-Sane et al., 2021; Zhang et al., 2020; Kim et al., 2023; Ajay et al., 2020; Pertsch et al., 2021). In practice, most recent methods cast subgoal choice as a *time-driven* decision. Midpoint priors between the current state and the goal are used only as training supervision (*midway-to-goal*, Chane-Sane et al., 2021). Fixed or skip-step schedules commit

to a new subgoal every $k$ steps or at coarse temporal resolutions (e.g., HIQL and hierarchical diffusion planners, Park et al., 2023; Janner et al., 2022; Ajay et al., 2023). In a similar spirit, fixed-horizon *skills* execute a $k$-step primitive or a stitched sub-trajectory before the high level revisits the choice (e.g., OPAL and diffusion-based sub-trajectory stitching, Ajay et al., 2020; Janner et al., 2022). A further strand overlays a short-window *value* criterion on top of the time window, selecting the highest-value candidate among states reachable in a few steps (e.g., ESD, Zhang et al., 2025). Despite their differences, these rules remain time/value–driven and are not bottleneck-aligned, thereby failing to exploit the valuable guidance signal the bottleneck can provide: where the agent must pass next. In short, we argue that high-level subgoals in offline GCRL should not be chosen via value estimates or time heuristics, but instead be derived from bottleneck states revealed by the Laplacian spectral structure of the offline data.

To address these issues, we advance a simple principle for hierarchical OGCRL: the optimal one-step subgoal is the next bottleneck. To identify bottlenecks from offline data, we first learn a Laplacian representation $\phi_\theta$ from replay so that states representation varies slowly within the same region while varies sharply at boundaries. Spectral clustering in this embedding yields region labels, Fig. 1 shows the resulting partition on `pointmaze-large`, where boundaries between colors coincide with hard-to-cross bottlenecks. We then identify trajectories from the offline dataset that cross these boundaries, and the intersects are defined as keypoints. Then the most representative KPs are automatically selected and a directed KP reachability graph $\mathcal{G}_{\mathrm{KP}}$ is constructed based on the selected KPs. At deployment, we restrict high-level choices to these KPs and use a pluggable low-level controller (e.g., Decision Diffuser or a lightweight MLP) to execute the short transitions between successive KPs. We name our method **BASS** (**B**ottleneck-**A**ware **S**pectral **S**ubgoaling).

**Contributions.** Our contributions form an integrated framework: (1) We introduce a bottleneck-guided criterion that ties subgoal selection to the next bottleneck, underpinned by theoretical analysis. (2) We develop a keypoint discovery method based on Laplacian spectral clustering to automatically extract bottleneck keypoints from offline datasets. (3) We design a hierarchical algorithm for OGCRL that plans by routing through these keypoints using a pluggable low-level controller. (4) Extensive experiments on diverse navigation and manipulation benchmarks from D4RL and OGBench demonstrate consistent bottleneck recovery and performance gains across varied data regimes.

## 2 PRELIMINARIES

**OGCRL and metastable bottlenecks.** We study offline goal-conditioned RL (OGCRL) where a policy $\pi(a \mid s, g)$ is learned from a fixed replay dataset. Long horizons with sparse rewards make progress hinge on planning rather than short-term value. In practice, most OGCRL tasks are *metastable*: the state space decomposes into regions that are easy to traverse, i.e., fast intra-region mixing, and these regions are connected only through a few hard-to-cross bottlenecks that are rarely crossed under the dataset-induced dynamics. Consequently, success to distant goals is governed by *whether the agent crosses the next bottleneck* rather than by taking a few more steps within the current region (see Fig. 1).

**Laplacian RL: what it is and why we use it.** Laplacian RL refers to representation-learning approaches that build on the low-frequency structure of the random-walk Laplacian induced by behavior dynamics (Wu et al., 2019; Wang et al., 2021a). The central idea is to encode long-horizon connectivity: states that are well connected in the data should have nearby embeddings, while states separated by bottlenecks should lie far apart.

The key property behind this is spectral: low-frequency eigencomponents of the Laplacian remain nearly constant within each metastable region but change sharply across bottlenecks. As a result, Euclidean distances in the learned embedding approximate diffusion-style reachability distances, stretching across bottlenecks and compressing within regions.

Formally, treating the dataset as a random walk with kernel $P$ over states, the random-walk Laplacian is $L_{\mathrm{rw}} = I - P$. Its low-frequency eigenvectors $\{e_i\}$ capture metastable structure. Mapping a state $s$ to its first $d$ non-trivial components,

$$\phi(s) = [e_1[s], \ldots, e_d[s]]^\top,$$

provides an embedding space aligned with the region–bottleneck topology.

In tabular settings one can obtain $\{e_i\}$ by eigendecomposition. In continuous state spaces, however, the Laplacian operator is infinite-dimensional, so we learn $\phi$ by minimizing a spectral graph-drawing objective with orthogonality constraints, estimated from mini-batches of transitions:

$$\min_{\{f_k\}_{k=1}^d} \sum_{k=1}^d \langle f_k, L f_k \rangle \quad \text{s.t.} \quad \langle f_j, f_k \rangle = \delta_{jk}, \ \langle f_k, \mathbf{1} \rangle = 0.$$

Earlier scalable formulations include the unconstrained graph-drawing objective (GDO) (Wu et al., 2019) and the generalized graph-drawing objective (GGDO) that breaks rotational symmetry at the cost of sensitive hyperparameters (Wang et al., 2021a). To avoid these issues, Proper Laplacian Representation Learning introduces the Augmented Lagrangian Laplacian Objective (ALLO) (Gomez et al., 2023), a min–max objective with stop-gradient asymmetry:

$$\max_\beta \ \min_{u \in \mathbb{R}^{d|S|}} \ \sum_{i=1}^d \langle u_i, L u_i \rangle \ + \ \sum_{j=1}^d \sum_{k=1}^j \beta_{jk} \langle u_j, u_k \rangle - \delta_{jk} \ + \ b \sum_{j=1}^d \sum_{k=1}^j \Big( \langle u_j, u_k \rangle - \delta_{jk} \Big)^2,$$

where $\beta_{jk}$ are dual variables, $b > 0$ is a barrier coefficient, and $\cdot$ denotes the stop-gradient operator. This objective uniquely recovers eigenvectors and their eigenvalues while removing untunable hyperparameters, we follow this when enforcing orthogonality and stability (details in Appendix).

# 3 THEORY IN A NUTSHELL: FROM LAPLACIAN SPECTRAL CLUSTERING TO OPTIMAL SUBGOALS

**Roadmap and intuition.** We study metastable environments where within-region movement is easy, while progress to distant goals is throttled by a few hard-to-cross bottlenecks. **Result I (bottleneck-guided subgoal optimality):** the next bottleneck is the optimal one-step subgoal. **Result II (spectral coverage):** when crossing a bottleneck is much harder and rarer than moving inside a region, and the learned Laplacian is accurate enough to reflect this, the low-frequency space provided by Laplacian representation could closely expose the true bottlenecks. Thereby, Laplacian spectral clustering recovers most bottlenecks with small error. **Combining I and II:** thus Laplacian spectral clustering can identify the

## 3.1 RESULT I: BOTTLENECK-GUIDED SUBGOAL OPTIMALITY

**Theorem 1** (Bottleneck-guidance optimality (condensed)). *Given a start $s \in R_{\mathrm{cur}}^\star$, a goal set $G \subseteq V \setminus R_{\mathrm{cur}}^\star$, and the next mandatory cross-section $\mathcal{B}^\star$ on any $s \to G$ path. Then*

$$\inf_{g \in V} \mathcal{J}(g) \ = \ T(s \to \mathcal{B}^\star) \ + \ \mathbb{E}_\xi\big[T(\xi \to G)\big] \ \pm \ O\big(t_{\mathrm{mix}}\big),$$

*where $\mathcal{J}(g) := T(s \to g) + T(g \to G)$ and $\xi \sim \mathrm{FirstHit}(s, \mathcal{B}^\star)$.*

Where $T(x \to A) := \mathbb{E}_x[\tau_A]$ is the expected hitting time, $t_{\mathrm{mix}}$ is the within-region mixing time of the reflected chain on $R_{\mathrm{cur}}^\star$, and $\mathrm{FirstHit}(s, \mathcal{B}^\star)$ is the first-hit distribution on $\mathcal{B}^\star$. Proof in Appendix.

**Design implication.** Pick the **next bottleneck** as the one-step subgoal. This is near-optimal whenever moving inside a region is easy and crossing the bottleneck is the main cost.

## 3.2 RESULT II: SPECTRAL CLUSTERING COVERAGE OF BOTTLENECKS

**Theorem 2** (Spectral clustering coverage of bottlenecks (condensed)). *Given a weighted, undirected graph $\mathcal{G} = (V, W)$ with random-walk kernel $P = D^{-1}W$, Laplacian $L = I - P$, $k$ metastable regions $\{R_i^\star\}_{i=1}^k$ satisfying $\Phi_{\mathrm{in}}(R_i^\star) \geq \alpha$ and $\Phi(R_i^\star \to R_j^\star) \leq \beta \ll \alpha$ for $i \neq j$, eigengap $\gamma = \lambda_{k+1} - \lambda_k > 0$, and an empirical Laplacian $\widehat{L}$ with deviation $\delta = \|\widehat{L} - L\|$. Let $\widehat{\mathcal{R}}$ be obtained by $k$-means on the row-normalized first $k$ eigenvectors of $\widehat{L}$. Then there exist $C_1, C_2, C_3 > 0$ such that*

$$\mathrm{MisVol} \ \leq \ C_1 \frac{\beta}{\alpha} \ + \ C_2 \frac{\delta}{\gamma}, \qquad \mathrm{Overlap}_\varepsilon \ \geq \ 1 \ - \ C_3 \, \mathrm{MisVol} \ - \ \mu\big(\mathcal{N}_\varepsilon(\partial \mathcal{R}^\star)\big).$$

Where $Q(S,T) = \sum_{u \in S, v \in T} \mu(u)P(u,v)$ is inter-set flow, $\Phi(S) = Q(S,S^c)/\mu(S)$ is conductance, $\Phi_{\text{in}}(R)$ is conductance of the reflected chain on $R$, $\text{MisVol} = \min_{\pi \in S_k} \sum_i \mu(\widehat{R}_{\pi(i)} \triangle R_i^\star)$ measures mis-clustered volume, $\text{Overlap}_\varepsilon = 1 - \mu(\mathcal{N}_\varepsilon(\partial\widehat{\mathcal{R}}) \triangle \mathcal{N}_\varepsilon(\partial\mathcal{R}^\star))/\mu(V)$ measures boundary overlap at tolerance $\varepsilon$, $\mu$ is the stationary distribution of $P$, and $\mathcal{N}_\varepsilon(\cdot)$ is an $\varepsilon$-neighborhood in the graph metric. Proof in Appendix.

**Design implication.** Learn a Laplacian embedding and cluster it. When (i) crossing a bottleneck is much rarer/harder than moving within a region, and (ii) the learned embedding faithfully reflects these transition patterns, the resulting cluster boundaries closely match the true bottlenecks.

> **Takeaway**
>
> The next bottleneck is the right one-step subgoal, and spectral clustering on a learned Laplacian can recover those bottlenecks under mild, data-driven conditions. Therefore, choosing subgoals at the discovered bottlenecks yields near-optimal plans with a small, interpretable gap.

## 4 METHOD

In this paper, we propose BASS (**B**ottleneck-**A**ware **S**pectral **S**ubgoaling) for OGCRL in environments where the state space consists of locally connected regions linked by a few hard-to-cross bottlenecks. Since crossing these bottlenecks dominates both time cost and failure risk, BASS follows a simple principle: find bottlenecks, then traverse bottlenecks. As shown in Fig. 2, we reveal bottlenecks from offline dataset via Laplacian spectral clustering and extract a dictionary of *keypoints* (KPs). Formally, we denote the state space by $\mathcal{S} \subseteq \mathbb{R}^D$ and learn a Laplacian encoder $\phi_\theta : \mathcal{S} \to \mathbb{R}^d$, with KPs given by $\hat{kp} \in \mathcal{V} = \hat{kp}_1, \ldots, \hat{kp}_M$.

At deployment, given $(s_0, g)$, we compute a KP routing over these keypoints, choose the next KP, and using a low-level controller to drive the system into that KP's acceptance region, a subset of state space decided by an distance predicate:

$$\mathcal{N}(\hat{kp}) = \{x \in \mathcal{S} : \text{dist}(x, \hat{kp}) \leq \varepsilon\},$$

with a single, task-agnostic tolerance $\varepsilon > 0$ per environment.

### 4.1 DISCOVER BOTTLENECKS

From offline dataset, we construct three artifacts that the deployment stage consumes: a Laplacian encoder $\phi_\theta$, a set of bottleneck keypoints $\mathcal{V}$, and a directed KP reachability graph $\mathcal{G}_{\text{KP}}$. We first learn $\phi_\theta$ to approximate the first non-zero $d$ ordered low-frequency eigenvectors of the random-walk Laplacian $L$ (Sec. 3), so that regions become nearly flat and bottlenecks become sharp in the embedding. Applying $K$-Means with a mildly over-segmented $K$ to $\phi_\theta(s)$ assigns region labels and reveals boundaries whose boundaries align with bottlenecks. Using these labels, we then sweep trajectories: whenever a transition $(s_t \to s_{t+1})$ crosses clusters and the new cluster persists for at least $\tau$ steps, we record $s_{t+1}$ as a crossing candidate, consolidating nearby candidates into one representative yields the KP set $\mathcal{V}$. Finally, we build the directed, unweighted graph $\mathcal{G}_{\text{KP}} = (\mathcal{V}, \mathcal{E})$ by connecting $i \to j$ if the dataset contains a short successful fragment such that, starting from the first hit of $\mathcal{N}(\hat{kp}_i)$, the trajectory first hits $\mathcal{N}(\hat{kp}_j)$ without entering any other KP region. Each edge thus encodes a single, data-supported hop between successive bottlenecks. We provide the pseudocode of this procedure in appendix A.

### 4.2 KP SEMANTICS AND ROUTING

Given $(s_0, g)$ and the offline artifacts $(\phi_\theta, \mathcal{V}, \mathcal{G}_{\text{KP}})$, we compute a shortest KP sequence $\hat{kp}_{i_0} \to \hat{kp}_{i_1} \to \cdots$ and return the *next* KP for execution. We drop the state coordinates that do not change at the KP and keep only those that do, which makes the KP easier to reuse on unseen goals. Formally, we represent each KP as

$$\text{KP} = (\mathcal{I}_\Delta, v_\Delta), \qquad \mathcal{I}_\Delta \subseteq \{1, \ldots, D\}, \quad v_\Delta \in \mathbb{R}^{|\mathcal{I}_\Delta|},$$

meaning that passing this KP deterministically sets $s[\mathcal{I}_\Delta] \leftarrow v_\Delta$ while leaving other coordinates unconstrained.

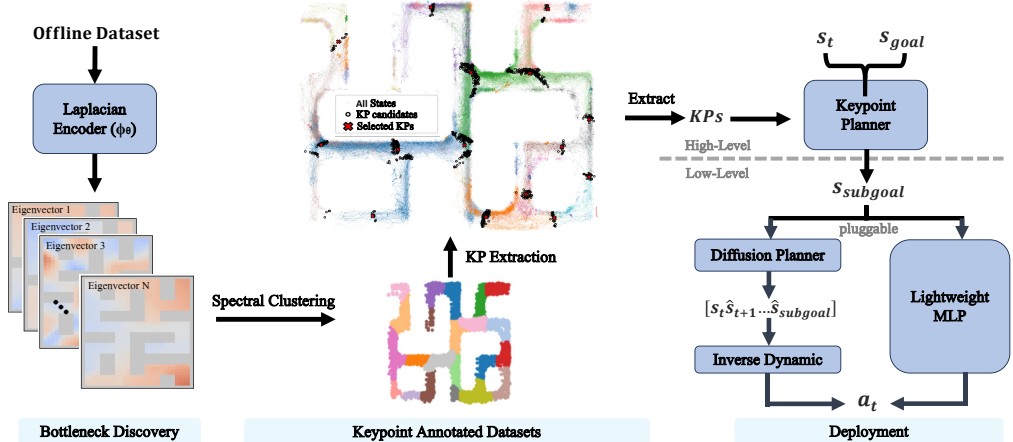

Figure 2: BASS overview. We learn a Laplacian encoder $\phi_\theta$ and apply spectral clustering to partition the state space into different regions, whose boundaries expose bottlenecks. We then identify trajectories from the offline dataset that cross these boundaries, and the intersects are defined as keypoints. Then the most representative KPs are automatically selected and a directed KP reachability graph $\mathcal{G}_{\mathrm{KP}}$ is constructed based on the selected KPs. Given $(s_t, g)$, a keypoint planner performs KP routing on $\mathcal{G}_{\mathrm{KP}}$, restricts choices to $\mathcal{V}$, and selects the next KP. A pluggable low-level controller, Decision Diffuser or a lightweight MLP, drives the system into the acceptance region $\mathcal{N}(\hat{kp})$, repeating this over KPs reaches the goal.

### 4.3 PLUGGABLE LOW-LEVEL CONTROLLERS

Once the next KP is selected, the controller only needs to drive the system into its acceptance region. We instantiate two interchangeable choices trained offline and selected by task demands at test time: (i) a Decision Diffuser that predicts a short state rollout $(s_t, \ldots, s_{t+k})$ and is paired with a lightweight inverse-dynamics MLP to recover actions from $(s_t, s_{t+1})$, and (ii) a Lightweight MLP that maps $(s_t, \text{next KP})$ directly to $a_t$ for fast inference. An optional keypoint regressor can predict an intermediate state $\tilde{s}_{t+k}$ to stabilize and shorten diffusion horizons. Inspired by the HIQL approach (Park et al., 2023), we train a small MLP keypoint regressor can predict an intermediate state $\tilde{s}_{t+k}$ to stabilize and shorten planning horizons.

When the diffusion route is optionally used, short trajectory segments are generated by simulating a reverse-time stochastic differential equation (SDE). Let $\mathbf{x}_t$ denote the vectorized planned trajectory at diffusion time $t$ and $q_t(\mathbf{x}_t)$ the diffused trajectory distribution. The reverse process follows

$$\mathrm{d}\mathbf{x}_t = \left[ f(t)\,\mathbf{x}_t - g(t)^2\,\nabla_\mathbf{x} \log q_t(\mathbf{x}_t) \right] \mathrm{d}t + g(t)\,\mathrm{d}\bar{\mathbf{w}}_t,$$

where the score $\nabla_\mathbf{x} \log q_t(\mathbf{x}_t)$ is approximated by the diffusion model's learned denoiser. We condition this process on the current state $s_t$ and an intermediate waypoint $\hat{s}_{t+k}$ at horizon $k$, then obtain $(s_t, \ldots, s_{t+k})$ and take $a_t = I(s_t, s_{t+1})$ via the inverse-dynamics model.

**Summary.** Offline we learn a Laplacian embedding, expose bottlenecks by clustering, extract KPs, and build $\mathcal{G}_{\mathrm{KP}}$. We route over KPs using sparse effects to minimize bottleneck crossings, and a pluggable controller executes each hop into the next acceptance region.

## 5 EXPERIMENTS

### 5.1 SETUP

**Environment** We evaluate on a unified suite of long-horizon, sparse-reward *offline* benchmarks spanning both navigation and manipulation, drawing from widely used D4RL and OGBench tasks. Concretely: **Maze2D/PointMaze/AntMaze/HumanoidMaze** require navigating complex maps (*umaze/medium/large/ultra/giant*) under sparse goal rewards, **FrankaKitchen** requires manipulating the scene by executing any four of seven object-centric skills to reach a target configuration. Our

datasets cover diverse regimes, including *play/diverse*, *stitch* and *partial* (test-time trajectories are longer than training snippets), and *explore* (low-quality data). In addition, we evaluate BASS on high-dimensional visual AntMaze variants; see Appendix for details and results.

**Baselines** We evaluate on two benchmark suites and align the baselines accordingly. On D4RL (Maze2D/AntMaze/HumanoidMaze/VisualAntmaze/FrankaKitchen), we compare against representative offline methods spanning four paradigms: goal-conditioned imitation **RvS-G** (Emmons et al., 2022), sequence models **Trajectory Transformer (TT)** (Janner et al., 2021), OGCRL methods **HIQL** (Park et al., 2023) and **ESD** (Zhang et al., 2025), and diffusion-based decision making **Diffusion-QL** (Wang et al., 2023c), **IDQL** (Hansen-Estruch et al., 2023), **Decision Diffuser (DD)** (Ajay et al., 2023), **Diffuser** (Janner et al., 2022), and **DIAR** (Park et al., 2024a). On OGBench, we report **HIQL** together with other OGCRL baselines enumerated in OGBench, including goal-conditioned behavioral cloning (GCBC) (Lynch et al., 2020), goal-conditioned implicit V-learning (GCIVL) (Park et al., 2024b), goal-conditioned implicit Q-learning (GCIQL) (Kostrikov et al., 2021; Zeng et al., 2023), Quasimetric RL (QRL) (Wang et al., 2023b), and Contrastive RL (CRL) (Eysenbach et al., 2022).

## 5.2 MAIN RESULTS

Table 1: **Performance comparison on D4RL** (success rate %, mean±std across 3 seeds, higher is better). We report AntMaze (Play/Diverse), FrankaKitchen, and Maze2D. Best in **bold**. "–" indicates not reported by prior work.

| Dataset | TT | RvS-G | HIQL | ESD | Diffusion-QL | IDQL | Diffuser | DD | DIAR | BASS (Ours) |
|---|---|---|---|---|---|---|---|---|---|---|
| *AntMaze (Play/Diverse)* | | | | | | | | | | |
| antmaze-umaze-play-v2 | **100.0** | 65.4 | 83.3 | 97.1±2.6 | 93.4±3.4 | 94.0 | 0.0 | 0.0 | – | 99.3±0.9 |
| antmaze-umaze-diverse-v2 | – | 60.9 | 85.4 | 92.9±4.2 | 66.2±8.6 | 80.2 | 0.0 | 0.0 | 88.8±1.5 | **98.0±1.6** |
| antmaze-medium-play-v2 | **100.0** | 58.1 | 86.8 | 90.8±6.4 | 76.6±10.8 | 84.5 | 0.0 | 0.0 | – | 98.0±0.0 |
| antmaze-medium-diverse-v2 | 93.3 | 57.3 | 84.1 | 88.3±6.0 | 78.6±10.3 | 84.8 | 0.0 | 0.0 | 68.2±6.7 | **96.7±0.9** |
| antmaze-large-play-v2 | 60.0 | 32.4 | 88.2 | 88.8±6.0 | 46.4±8.3 | 63.5 | 0.0 | 0.0 | – | **96.0±1.6** |
| antmaze-large-diverse-v2 | 66.7 | 36.9 | 86.1 | 87.9±5.0 | 56.6±7.6 | 67.9 | 0.0 | 0.0 | 60.6±2.4 | **98.7±1.9** |
| antmaze-ultra-play-v2 | 33.3 | – | 52.9 | 56.7±9.1 | – | – | 0.0 | 0.0 | – | **97.3±0.3** |
| antmaze-ultra-diverse-v2 | 20.0 | – | 39.2 | 55.8±11.3 | – | – | 0.0 | 0.0 | – | **88.0±1.6** |
| **Average (AntMaze)** | – | – | 75.7 | 82.3 | 69.6 | – | 0.0 | 0.0 | – | **96.5** |
| *FrankaKitchen* | | | | | | | | | | |
| kitchen-partial-v0 | – | – | 65.0 | 69.8±2.1 | 60.5±6.9 | – | 56.2 | 57.0 | 63.3±0.9 | **83.3±4.9** |
| kitchen-mixed-v0 | – | – | 67.7 | 67.1±5.0 | 62.6±5.1 | – | 50.0 | 65.0 | 60.8±1.4 | **86.0±2.8** |
| **Average (Kitchen)** | – | – | 66.4 | 68.5 | 61.2 | – | 53.1 | 61.0 | 62.5 | **84.5** |
| *Maze2D* | | | | | | | | | | |
| maze2d-large-v1 | – | – | – | – | – | 90.1 | 123.0 | – | **200.3±3.4** | 189.3±6.2 |

Table 2: **Performance comparison on OGBench** (success rate %, mean±std across 3 seeds). Baselines come from the OGBench reports. Best in **bold**.

| Dataset | GCBC | GCIVL | GCIQL | QRL | CRL | HIQL | BASS (Ours) |
|---|---|---|---|---|---|---|---|
| *PointMaze* | | | | | | | |
| pointmaze-large-navigate-v0 | 29 ± 6 | 45 ± 5 | 34 ± 3 | 86 ± 9 | 39 ± 7 | 58 ± 5 | **97.3±1.2** |
| pointmaze-giant-navigate-v0 | 1 ± 2 | 0 ± 0 | 0 ± 0 | 68 ± 7 | 27 ± 10 | 46 ± 9 | **88.0±6.0** |
| pointmaze-teleport-navigate-v0 | 25 ± 3 | **45±3** | 24 ± 7 | 4 ± 4 | 24 ± 6 | 18 ± 4 | 22.0 ± 4.0 |
| pointmaze-large-stitch-v0 | 7 ± 5 | 12 ± 6 | 31 ± 2 | 84 ± 15 | 0 ± 0 | 13 ± 6 | **99.3±1.2** |
| pointmaze-giant-stitch-v0 | 0 ± 0 | 0 ± 0 | 0 ± 0 | 50 ± 8 | 0 ± 0 | 0 ± 0 | **85.3±3.1** |
| pointmaze-teleport-stitch-v0 | 31 ± 9 | **44±2** | 25 ± 3 | 9 ± 5 | 4 ± 3 | 34 ± 4 | 42.0 ± 14.0 |
| *AntMaze (OGBench variants)* | | | | | | | |
| antmaze-large-stitch-v0 | 3 ± 3 | 18 ± 2 | 7 ± 2 | 18 ± 2 | 11 ± 2 | 67 ± 5 | **81.0±7.0** |
| antmaze-giant-stitch-v0 | 0 ± 0 | 0 ± 0 | 0 ± 0 | 0 ± 0 | 0 ± 0 | 2 ± 2 | **71.3±7.0** |
| antmaze-large-explore-v0 | 0 ± 0 | 10 ± 3 | 0 ± 0 | 0 ± 0 | 0 ± 0 | 4 ± 5 | **72.7±1.2** |
| *HumanoidMaze* | | | | | | | |
| humanoidmaze-large-navigate-v0 | 1 ± 0 | 2 ± 1 | 2 ± 1 | 5 ± 1 | 24 ± 4 | 49 ± 4 | **57.3±3.1** |
| humanoidmaze-giant-navigate-v0 | 0 ± 0 | 0 ± 0 | 0 ± 0 | 1 ± 0 | 3 ± 2 | 12 ± 4 | **62.0±9.2** |
| humanoidmaze-large-stitch-v0 | 6 ± 3 | 1 ± 1 | 0 ± 0 | 3 ± 1 | 4 ± 1 | 28 ± 3 | **45.3±3.1** |
| humanoidmaze-giant-stitch-v0 | 0 ± 0 | 0 ± 0 | 0 ± 0 | 0 ± 0 | 0 ± 0 | 3 ± 2 | **55.3±3.1** |
| *Visual Antmaze* | | | | | | | |
| visual-antmaze-large-navigate-v0 | 4 ± 0 | 5 ± 1 | 4 ± 1 | 0 ± 0 | **84±1** | 53 ± 9 | 78.7±2.3 |
| visual-antmaze-large-stitch-v0 | 24 ± 3 | 1 ± 1 | 0 ± 0 | 1 ± 1 | 11 ± 3 | 28 ± 2 | **68.0±2.0** |

Across a wide variety of navigation and manipulation tasks, scales, and dataset regimes, our method achieves *consistently higher* success than prior work, highlighting the advantage of using bottlenecks as subgoals. We also try to explain where the gains arise: (i) in precision-sensitive settings (e.g., *AntMaze* corners where high-DoF agents often stumble, *Kitchen* grasps that easily miss), placing subgoals *on* bottlenecks lets the agent finely adjust to enter the KP acceptance region, (ii) in time-limit–prone layouts (*PointMaze/HumanoidMaze* with many detour traps), KP routing finds short KP chains, and bottleneck-anchored subgoals steer the agent onto the correct corridor early, avoiding costly backtracking.

## 5.3 Generalization Across Environments

We evaluate generalization along two axes, organized from upper to lower levels in our hierarchy: **(G1)** High-level transfer: swapping keypoint graphs across domains, and **(G2)** Low-level transfer: controller across AntMaze scales. For (G1), we swap keypoint graphs among three datasets with similar state space, including *PointMaze-large-Stitch*, *AntMaze-large-Stitch*, and *AntMaze-large-Explore*, and test them across domains. For (G2), we take a diffuser-based low-level controller trained on AntMaze-Large-Play and transfer it to other AntMaze scales. Results are summarized in Tab. 3 and Tab. 4.

Table 3: (G1) Cross-domain transfer between PointMaze and AntMaze using swapped keypoint graphs. Rows: KP source; Columns: test environment.

| KP source → Test env | Point-Stitch | Ant-Stitch | Ant-Explore |
|---|---|---|---|
| Point-Stitch | **99.3±1.2** | **83.3±6.4** | **87.3±3.1** |
| AntMaze-Stitch | **99.3±1.2** | $81.0 \pm 7.0$ | $65.3 \pm 5.0$ |
| Ant-Explore | $98.7 \pm 1.2$ | $66.7 \pm 3.1$ | $72.7 \pm 1.2$ |

**(G1) High-level transfer: swapping keypoint graphs across domains.** Tab. 3 shows that exchanging the keypoint graph among the three tasks on the same map does not significantly hurt performance. We emphasize that this is a diagnostic experiment: for each target environment, the in-domain BASS row serves as the reference, and the other rows simply reuse the same policy with a swapped keypoint graph. Our interpretation is that the KP graph captures the topological backbone of the state space, critical corridors and bottlenecks, rather than the shaping of task-specific rewards. As a result, planning on this graph remains valid even when the task or data source differs, yielding stable success rates.

More interestingly, tranfering PointMaze KPs to AntMaze leads to higher success than native AntMaze KPs. We hypothesize that PointMaze's simpler point-mass dynamics produce offline data with smoother intra-region transitions and cleaner inter-region boundaries. This makes graph construction and the Laplacian-based representation more faithful to true connectivity and bottlenecks according to Theorm 1. When reused in AntMaze, the upper level then proposes subgoals that better align with the topological structure of the maze, while the lower level absorbs the actuation complexity of the ant. This suggests a promising direction: learn KPs in simple domains with rich coverage, and transfer them to more complex domains that share a similar state space and transition structure.

Table 4: (G2) Frozen controller transferred across map scales (success %).

| Source map → Target map | Umaze | Medium | Large | Ultra |
|---|---|---|---|---|
| Large | **98.7±1.2** | **96.7±2.3** | **96.0±1.6** | **96.7±1.2** |

**(G2) Low-level transfer: controller across AntMaze scales.** Tab. 4 demonstrates that a diffuser planner trained on AntMaze-Large-Play generalizes strongly to other map scales when paired with each target's own KPs. Although global layouts differ, the controller receives short-horizon subgoals from the upper level and only needs to execute local, easy-to-learn skills including move-to-subgoal and pass-corridor. This decomposition makes the controller largely insensitive to global map differences and encourages robust, reusable primitives. In other words, choosing bottlenecks as subgoals

provides near-optimal guidance, reducing the lower level to a simpler, transferable control task. This observation supports Theorem 2.

## 5.4 ABLATION STUDIES

**Ablation of the bottleneck-guided subgoals** We ablate the high-level subgoal selector. Our method identifies subgoals at bottlenecks via Laplacian spectral clustering. As a drop-in replacement, we use the common time-based rule from hierarchical offline goal-conditioned RL (Chane-Sane et al., 2021; Zhang et al., 2020; Kim et al., 2023; Ajay et al., 2020; Pertsch et al., 2021): following HIQL, every fixed horizon `way_steps` we choose the state with the highest value as the subgoal. We test two typical `way_steps`, 25 (HIQL default) and 5. Tab. 5 shows results on *antmaze-large-play/diverse-v2*. Replacing bottleneck subgoals with the time-based HIQL variant causes substantial drops, especially at the short horizon. This indicates that bottleneck-guided subgoals are the primary driver of our gains. The evidence also supports **Theorem 1**, which predicts that bottlenecks are near-optimal subgoals under our assumptions, whereas short-horizon value peaks can be myopic and ignore global connectivity.

Table 5: Ablation of the bottleneck-guided subgoals on *antmaze-large-play/diverse-v2*

| Setting | Large-Play-v2 | Large-Diverse-v2 |
|---|---|---|
| BASS (ours) | **98.0 ± 1.6** | **98.7 ± 1.9** |
| BASS w/ HIQL Keypoint & way_step=25 | 83.3 ± 3.1 | 84.0 ± 5.3 |
| BASS w/ HIQL Keypoint & way_step=5 | 18.7 ± 2.5 | 22.7 ± 0.9 |

**Ablation of the Number of Clusters $K$.** To study how the number of clusters $K$ in Laplacian spectral clustering affects performance, we vary $K$ on four representative environments, the results are shown in Table 6 and 7. Across these tasks we observe a consistent pattern: very small $K$ yields overly coarse partitions that under-detect bottlenecks and hurt performance; there is a broad plateau of $K$ where performance is stable and often matches or even exceeds the numbers in the main tables; and only a few environments does very large $K$ can slightly reduce performance by introducing unnecessary path complexity. This trend supports **Theorem 2** that the operative criterion for Laplacian spectral clustering here is to cover bottlenecks.

## 5.5 VISUALIZATION

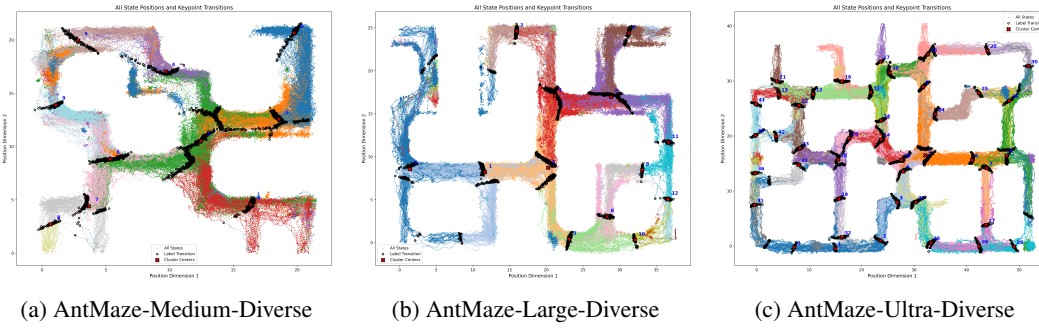

(a) AntMaze-Medium-Diverse      (b) AntMaze-Large-Diverse      (c) AntMaze-Ultra-Diverse

Figure 3: Trajectories and keypoints in three AntMaze layouts. Colors indicate metastable regions, black dots denote transition keypoints, red crosses mark selected KPs.

In Fig. 3, colors delineate metastable regions, black dots mark transitions across bottlenecks, and red crosses are the KPs used by the high-level policy. Keypoints concentrate at intersections precisely where conductance is low and paths must cross—validating that spectral clustering recovers bottlenecks. Aligning subgoals with these bottlenecks simplifies the task: high-level routing picks short KP chains, while low-level only needs to enter the next KP's acceptance region. This bottleneck-guided decomposition explains the robust gains observed across scales and datasets. We also hand-annotate

oracle keypoints on antmaze-large-diverse at the centers of required corners and visually compare the resulting trajectories with those from BASS; see App. C.

## 6 RELATED WORK

**Goal Conditioned Offline RL** One of the mainstream OGCRL approach defines subgoals as midpoints between the current state $s$ and the goal $g$, incorporating them as priors during training but leaving them unused during testing (Chane-Sane et al., 2021). This method enhances learning by introducing additional supervision. Another approach (Zhang et al., 2020) constrains subgoals within a k-step neighborhood to maintain local feasibility within a limited horizon. In addition, graph-based planning methods also support GCRL. For instance, (Kim et al., 2023) treats subgoals as nodes in a graph, where edges represent advantages between them and are used for path planning during inference. This combines goal-conditioned policies with graph-based reasoning to facilitate task completion. Meanwhile, diffusion-based methods have been used to output low-level control signals or to plan long-horizon trajectories (e.g., Decision Diffuser and Diffuser) and can be plugged into these frameworks (Ajay et al., 2022; Janner et al., 2022). Besides, hierarchical planning approaches explore subgoal generation using graphs or models. (Fang et al., 2023) predicts subgoals autoregressively with latent space representations of future states, while (Li et al., 2022) generates subgoals at regular intervals, similar to autonomous vehicle navigation systems predicting future keypoints. $CE^2$ (Duan et al., 2024) leverages cluster boundaries in a learned latent space for online goal-directed exploration, while our focus is on offline planning and subgoal selection.

Quasimetric RL (QRL) (Wang et al., 2023a) learns a temporal-distance function and uses it to regularize value learning and planning. HILP (Park et al., 2024c) plans in a temporal latent space and chooses subgoals as evenly spaced latent states along a trajectory. Graph-Assisted Stitching (GAS) (Baek et al., 2025) formulates subgoal selection as graph search in a temporal-distance representation, emphasizing micro-level trajectory stitching across offline data. In contrast, BASS discovers macro-level bottleneck keypoints via Laplacian structure.

**Laplacian Representation** In Laplacian representation learning for reinforcement learning (RL), early work (Mahadevan & Maggioni, 2007) introduced Proto-Value Functions (PVF), leveraging random-walk Laplacian eigenvectors for state representation. (Wu et al., 2018) expanded this by proposing a Graph Drawing Objective (GDO) for large state spaces, but it struggled with eigenvector rotations and hyperparameter tuning. (Wang et al., 2021b) introduced the Generalized Graph Drawing Objective (GGDO), which improved upon GDO by breaking symmetry, but still faced hyperparameter sensitivity and failed to recover eigenvalues accurately. (Gomez et al., 2023) introduced the Augmented Lagrangian Laplacian Objective (ALLO), which addresses the shortcomings of GDO and GGDO. ALLO eliminates hyperparameter dependence, accurately recovers both eigenvectors and eigenvalues, and provides more stable and accurate results across environments, advancing the field significantly. In addition, (Klissarov & Machado, 2023) used Laplacian representations to improve exploration. By contrast, our work uses Laplacian structure to build a bottleneck keypoint graph for long-horizon offline goal-conditioned decision-making, focusing on discovering semantic bottlenecks and routing through them rather than on exploration per se.

## 7 CONCLUSIONS

We reframed offline goal-conditioned RL as routing through metastable regions connected by a few hard-to-cross bottlenecks. Our principle is simple: the near-optimal one-step subgoal is the next bottleneck. We operationalize this by learning a Laplacian representation from offline data, applying spectral clustering to expose bottlenecks, extracting keypoints (KPs) at the crossings, and planning with a lightweight, dynamics-agnostic BFS over the KP graph. A pluggable low-level controller, either a Decision Diffuser or a lightweight MLP, then drives the system into each KP's acceptance region.

Theory establishes subgoal optimality (Theorem 1) and boundary recovery (Theorem 2), implying near-optimal routing. Experiments on D4RL and OGBench achieve state-of-the-art success and generalize across controllers, domains, and scales, including KP-graph swapping (G1) and controller transfer across AntMaze scales (G2).

ETHICS STATEMENT

This work studies offline goal-conditioned RL with bottleneck-guided subgoals on public benchmarks (D4RL and OGBench). No human subjects or private data are used; all datasets and libraries follow their original licenses. Potential risks include unintended behaviors when policies are deployed out of distribution, bias inherited from offline logs, and additional compute/energy costs. We do *not* deploy to real robots; all results are in simulation. We recommend human oversight, safety constraints, and compliance review for any downstream, high-stakes use.

REPRODUCIBILITY STATEMENT

We provide implementation details in the appendix, like Laplacian training objective and optimizer settings, BFS routing, and low-level controller configurations. We report mean±std over three seeds. Code will be released after the camera-ready version is finalized.

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

---

**Algorithm 1** Bottleneck keypoint discovery

---

**Require:** Offline dataset $\mathcal{D}_{\text{off}} = \{\mathbf{h}_i\}$, number of clusters $K$, boundary persistence $\tau$
**Ensure:** Keypoint set KPs
 1: **// Laplacian representation and spectral clustering**
 2: Train a Laplacian encoder $\phi$ on states $\{s \mid (s, \cdot) \in \mathcal{D}_{\text{off}}\}$ and obtain embeddings $z = \phi(s)$.
 3: Run $K$-means with $K$ clusters on $\{z\}$; let $c(s) \in \{1, \ldots, K\}$ be the cluster label of state $s$.
 4: **// Collect boundary samples between clusters**
 5: Initialize boundary buffer $\mathcal{B} \leftarrow \emptyset$.
 6: **for** each trajectory $h = (s_0, \ldots, s_T)$ in $\mathcal{D}_{\text{off}}$ **do**
 7:     **for** $t = 0, \ldots, T - \tau_b - 1$ **do**
 8:         **if** $c(s_t) \neq c(s_{t+1})$ **and** $c(s_{t+1}) = \cdots = c(s_{t+\tau_b})$ **then**
 9:             Append $s_{t+1}$ to $\mathcal{B}$                     ▷ candidate boundary state
10:         **end if**
11:     **end for**
12: **end for**
13: **// Compress boundary samples into keypoints**
14: Initialize keypoint set KPs $\leftarrow \emptyset$.
15: Group boundary samples in $\mathcal{B}$ into small neighborhoods of nearby states and compute a representative center $\mu_\ell$ for each group.
16: **for** each center $\mu_\ell$ **do**
17:     Construct a keypoint $\text{KP}_\ell = (I_\Delta, v_\Delta)$ from $\mu_\ell$ as described in Sec. 4.2.
18:     Add $\text{KP}_\ell$ to KPs.
19: **end for**
20: **return** KPs

---

# A   PSEUDO-CODE FOR BOTTLENECK DISCOVERY

# B   ABLATION OF THE NUMBER OF CLUSTERS K

Table 6: The performance of our method with different numbers of clusters on *antmaze-giant-stitch* and *pointmaze-giant-stitch*

| **keypoints** | 30 | 32 | 34 | 36 | 38 | 40 | 42 | 44 |
|---|---|---|---|---|---|---|---|---|
|  | 46 | 48 | 50 | 52 | 54 | 56 | 58 | |
| **antmaze-** | $0.0 \pm 0.0$ | $13.3 \pm 2.3$ | $20.0 \pm 7.0$ | $11.3 \pm 3.1$ | $40.0 \pm 8.0$ | $53.3 \pm 3.1$ | $40.7 \pm 4.2$ | $60.0 \pm 9.2$ |
| **giant-stitch** | $62.0 \pm 6.0$ | $68.0 \pm 3.5$ | $71.3 \pm 7.0$ | $63.3 \pm 2.3$ | $66.7 \pm 2.3$ | $66.7 \pm 5.0$ | $61.3 \pm 3.1$ | |
| **pointmaze-** | $92.0 \pm 2.0$ | $84.7 \pm 1.2$ | $80.7 \pm 3.1$ | $86.7 \pm 3.1$ | $90.0 \pm 5.3$ | $81.3 \pm 1.2$ | $78.7 \pm 4.2$ | $84.7 \pm 5.1$ |
| **giant-stitch** | $80.0 \pm 3.5$ | $88.7 \pm 2.3$ | $85.3 \pm 3.1$ | $83.3 \pm 6.1$ | $85.3 \pm 6.1$ | $88.7 \pm 6.1$ | $84.7 \pm 3.1$ | |

# C   TRAJECTORY VISUALIZATION AND COMPARISON WITH EXPERT HAND-ANNOTATED TRAJECTORIES

# D   IMPLEMENTATION DETAILS OF THE LAPLACIAN LOSS

In our framework, the Laplacian representation is learned by minimizing a loss function that creates a feature space reflecting the temporal connectivity of the state space. In this representation space, states that require many transitions to connect (i.e., have long transition durations) are far apart, while states that are easily reachable (i.e., with short transition periods) are embedded close together. Such a design not only naturally measures transition difficulty but also highlights bottlenecks and regions where rapid changes in the learned representation indicate potential sub-task boundaries. These boundaries manifest as clustering limits where keypoints are more likely to occur.

Table 7: The performance of our method with different numbers of clusters on *antmaze-large-play* and *pointmaze-large-stitch*.

| keypoints | 10 | 15 | 20 | 22 | 24 | 26 | 28 |
|---|---|---|---|---|---|---|---|
| | 30 | 32 | 34 | 36 | 38 | 40 | 42 |
| | 44 | 46 | 48 | 50 | | | |
| **antmaze-large-play** | $32.0 \pm 4.0$ | $15.3 \pm 7.0$ | $96.0 \pm 1.6$ | $93.3 \pm 3.1$ | $94.7 \pm 3.1$ | $98.0 \pm 2.0$ | $91.3 \pm 3.1$ |
| | $91.3 \pm 1.2$ | $95.3 \pm 1.9$ | $\mathbf{98.0 \pm 0.0}$ | $92.0 \pm 1.6$ | $96.7 \pm 0.9$ | $90.7 \pm 3.4$ | $87.3 \pm 5.0$ |
| | $94.0 \pm 3.3$ | $89.3 \pm 8.1$ | $90.0 \pm 3.3$ | $79.3 \pm 4.1$ | | | |
| **pointmaze-large-stitch** | $98.0 \pm 1.6$ | $100.0 \pm 0.0$ | $100.0 \pm 0.0$ | $100.0 \pm 0.0$ | $99.3 \pm 0.9$ | $100.0 \pm 0.0$ | $100.0 \pm 0.0$ |
| | $98.0 \pm 3.5$ | $100.0 \pm 0.0$ | $100.0 \pm 0.0$ | $100.0 \pm 0.0$ | $96.0 \pm 2.0$ | $100.0 \pm 0.0$ | $100.0 \pm 0.0$ |
| | $100.0 \pm 0.0$ | $100.0 \pm 0.0$ | $100.0 \pm 0.0$ | $96.7 \pm 1.2$ | | | |

Table 8: Average evaluation steps for HIQL and our method across different environments.

| Dataset | HIQL Steps | BASS (ours) Steps |
|---|---|---|
| **antmaze-giant-stitch-v0** | 997.50 | 864.17 |
| **antmaze-large-stitch-v0** | 640.87 | 547.28 |
| **pointmaze-large-stitch-v0** | 905.04 | 265.33 |
| **humanoidmaze-large-navigate-v0** | 1667.65 | 1652.34 |
| **humanoidmaze-large-stitch-v0** | 1808.40 | 1717.22 |
| **humanoidmaze-giant-navigate-v0** | 3900.32 | 3287.94 |
| **humanoidmaze-giant-stitch-v0** | 3978.45 | 3319.76 |

Our implementation fully adheres to (Gomez et al., 2023), which training proceeds through four high-level stages:

## D.1 DATA SAMPLING

- **Graph-Drawing (Primal) Pairs:** From the replay buffer or trajectory dataset, randomly sample state-transition pairs $(s_t, s_{t+n})$. These capture the temporal *difficulty* of moving from $s_t$ to $s_{t+n}$ over a fixed (or randomly chosen) horizon $n$, exactly as in the classical Laplacian spectral objective.

- **Orthogonality (Constraint) Batches:** Independently sample two small batches of states $\{s_i^1\}$ and $\{s_i^2\}$. These are not paired but serve to enforce near-orthogonality between different embedding dimensions, consistent with the proper Laplacian constraint.

## D.2 REPRESENTATION ENCODING

A single encoder network $\phi_\theta$ maps each sampled state into a $d$-dimensional embedding:

$$u = \phi_\theta(s) \in \mathbb{R}^d.$$

- $\phi_\theta(s_t)$ and $\phi_\theta(s_{t+k})$ are used to compute the graph-drawing loss, matching the $\langle u, Lu \rangle$ term of the proper Laplacian.

- $\phi_\theta(s_i^1)$ and $\phi_\theta(s_i^2)$ are used to compute the orthogonality error matrix, implementing the $u^T u = I$ constraint softly.

## D.3 LOSS CONSTRUCTION

We combine three terms into a single augmented Lagrangian that exactly mirrors the proper Laplacian objective:

$$\mathcal{L}_{\text{total}} = \mathcal{L}_{\text{graph}} + \mathcal{L}_{\text{dual}} + \mathcal{L}_{\text{barrier}}.$$

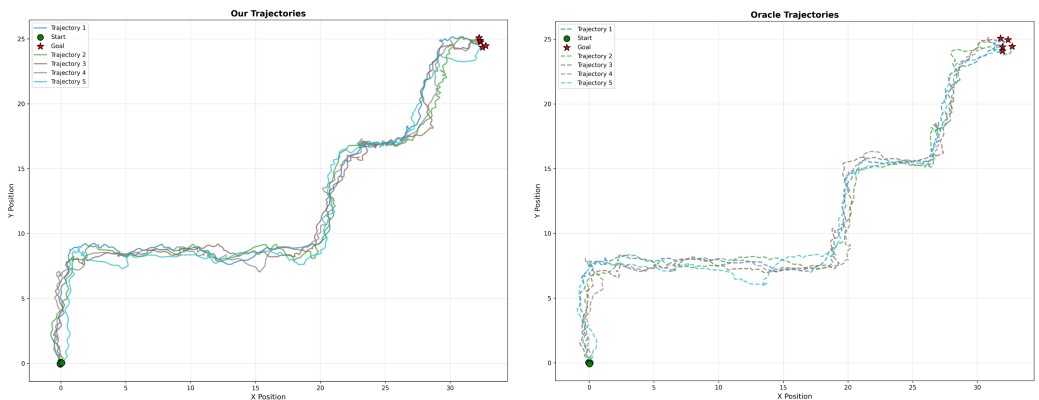

Figure 4: Comparison between our trajectories and oracle trajectories.

- **Graph-Drawing Term:**

$$\mathcal{L}_{\text{graph}} = \sum_{i=1}^{d} (u_t^i - u_{t+k}^i)^2 \times \text{coeff}_i,$$

which exactly implements the $\langle u, Lu \rangle$ spectral penalty.

- **Linear Lagrangian (Dual) Term:**

$$\mathcal{L}_{\text{dual}} = \sum_{j \geq k} \beta_{jk} \langle u_j, \text{stopgrad}(u_k) \rangle,$$

with dual variables $\beta_{jk}$ enforcing the orthogonality constraints in the augmented Lagrangian sense.

- **Quadratic Barrier Penalty:**

$$\mathcal{L}_{\text{barrier}} = b \sum_{j \geq k} (\langle u_j, \text{stopgrad}(u_k) \rangle - \delta_{jk})^2,$$

softly enforcing $u^T u = I$, consistent with the proper Laplacian spectral formulation.

## D.4 JOINT OPTIMIZATION WITH ALTERNATING UPDATES

1. **Encoder Update:** With $\beta$ and $b$ fixed, minimize $\mathcal{L}_{\text{total}}$ w.r.t. $\theta$, exactly following the proper spectral embedding procedure.

2. **Dual Variables Update:** With $\theta$ fixed, perform a projected gradient *ascent* step on $\beta$ using current orthogonality errors, corresponding to the update of Lagrange multipliers.

3. **Barrier Scheduling:** Increase $b$ over training—on a schedule or when constraint violations persist—to maintain the strength of the barrier term, as in augmented Lagrangian methods.

## D.5 SUMMARY

By strictly following the classical Laplacian spectral graph objective and its augmented Lagrangian relaxation—combining

1. a graph-drawing term preserving *transition difficulty*,

2. a linear Lagrangian term enforcing orthonormality,

3. a quadratic barrier penalty for soft constraints,

and by alternating minimization for the encoder with maximization for the duals, we obtain a proper Laplacian embedding that faithfully preserves temporal connectivity and yields disentangled, stable representations for downstream keypoint detection and hierarchical control.

# E   RESIDUAL-STATE BFS FOR KP ROUTING

In the main text (§4.2), we represent a keypoint (KP) as

$$\mathrm{KP} = (\mathcal{I}_\Delta, v_\Delta), \qquad \mathcal{I}_\Delta \subseteq \{1, \ldots, D\}, \quad v_\Delta \in \mathbb{R}^{|\mathcal{I}_\Delta|},$$

which deterministically sets the coordinates in $\mathcal{I}_\Delta$ to $v_\Delta$. Given a start–goal pair $(s_0, g)$, we plan only over coordinates that differ from the goal:

$$\mathcal{R}_0 = \{\, i : \, s_0[i] \neq g[i] \,\}, \qquad q = |\mathcal{R}_0|.$$

For each KP, we keep only its goal-aligned footprint

$$F(\mathrm{KP}; g) = \{\, i \in \mathcal{I}_\Delta : \, v_\Delta[i] = g[i]\}.$$

**Routing.** We perform breadth-first search *only over KPs that can change at least one currently unsatisfied coordinate* (i.e., $F(\mathrm{KP}; g) \cap \mathcal{R} \neq \varnothing$), and apply light pruning: skip no-op KPs (no residual coverage), de-duplicate visited residual sets, and optionally prioritize candidates by $|F \cap \mathcal{R}|$ to reduce expansions while preserving shortest-path optimality.

**Complexity and scale.** Let $d$ denote the average number of KPs whose footprints intersect the current residual (average branching factor). In the worst case, the number of residual sets visited is bounded by $2^q$, and each expansion considers $O(d)$ candidates:

$$O(d \cdot 2^q) \text{ time}, \qquad O(2^q) \text{ space}.$$

In our OGCRL settings, both quantities are *very small* in practice (empirically $d < 5$, $q < 10$), making runtime acceptable.

# F   LOW-LEVEL STRATEGY (PLUGGABLE)

Our low level is *modular* and exposes a unified interface

$$a_t = \mathrm{LOWLEVEL}(s_t, \tilde{g}_t; \eta),$$

where $\tilde{g}_t$ is the high-level keypoint–guided mid-goal and $\eta$ are backend hyperparameters. We run in a receding horizon: compute $a_t$ from $(s_t, \tilde{g}_t)$, step the env to get $s_{t+1}$, and repeat.

## F.1   KEYPOINT-CONDITIONED $k$-STEP STATE PREDICTION

Because time-to-reach a keypoint is uncertain while planners often assume a fixed horizon $k$, we first predict a *$k$-step target state* $s_{t+k}$ conditioned on the current state and the selected keypoint:

$$s_{t+k} = \pi_\omega(s_t, k_i),$$

where $k_i$ is the keypoint selected by the high level. Concretely:

- **Inputs.** $(s_t, k_i)$.
- **Objective.** A value model $V_\phi(s, k)$ provides HIQL-style supervision to train $\pi_\omega$ so that the predicted $s_{t+k}$ maximizes the expected keypoint-conditioned return over $k$ steps.
- **Output.** $s_{t+k}$, which anchors a short-horizon plan.

This $k$-step target is then consumed by one of two interchangeable backends.

## F.2   BACKEND A: SHORT-HORIZON DIFFUSION PLANNER (DECISION DIFFUSER)

**Conditioning.** Generate a $k$-step local plan from $s_t$ to the target $s_{t+k}$ by conditional diffusion, using $(s_t, s_{t+k})$ (or $(s_t, \tilde{g}_t)$ if planning in action space) as conditioning signals.

**Sampling.** A time-indexed network $\epsilon_\theta(\cdot, t)$ approximates the reverse score to produce a smooth trajectory $\{s_t, \ldots, s_{t+k}\}$ with a small number of reverse steps.

**State→Action.** If planning in *state space*, actions are recovered via a lightweight inverse-dynamics MLP $I_\zeta$:

$$a_t = I_\zeta(s_t, s_{t+1}),$$

trained with MSE on offline transitions. If planning directly in *action space*, $I_\zeta$ is not used.

Table 9: Hyperparameters

| Hyperparameter | Value & Specifics |
|---|---|
| d (embedding dim.) | 21 = 1 zero-eigenvector + 20 low-frequency eigenvectors |
| other Laplacian representation params | follow Gómez et al. (2023) settings |
| k (intermediate horizon) | 5 for AntMaze, 25 for Kitchen |
| T (diffusion steps) | 5 for AntMaze, 50 for Kitchen |
| Diffusion model | DiT with hidden_dim=384, nhead=8, layers=3 |
| Optimizer (diffusion) | AdamW with lr=$2 \times 10^{-4}$ |
| Optimizer (ivdm) | weight_decay=$1 \times 10^{-5}$ |
| Inverse dynamics (hidden_size) | MLP hidden_size=256, optimizer Adam lr=$2 \times 10^{-4}$ |

### F.3 BACKEND B: GOAL-CONDITIONED REACTIVE CONTROLLER (GC-MLP)

**Inputs.** Concatenate the current state and subgoal: $x_t = [s_t, \tilde{g}_t]$ (or $[s_t, s_{t+k}]$). A small MLP $f_\psi$ outputs $a_t = f_\psi(x_t)$.

**Training: IQL Objective.** We train the GC-MLP with the IQL loss, together with a value network $V_\phi$ and a critic $Q_\theta$:

$$\mathcal{L}_Q(\theta) = \mathbb{E}_{(s,a,r,s') \sim \mathcal{D}} \Big[ \big( Q_\theta(s,a) - \big( r + \gamma V_\phi(s') \big) \big)^2 \Big], \tag{1}$$

$$\mathcal{L}_V(\phi) = \mathbb{E}_{(s,a) \sim \mathcal{D}} \Big[ \rho_\tau \big( Q_\theta(s,a) - V_\phi(s) \big) \Big], \quad \rho_\tau(\delta) = |\tau - \mathbb{1}\{\delta < 0\}| \, \delta^2, \tag{2}$$

$$\mathcal{L}_\pi(\psi) = \mathbb{E}_{(s,a) \sim \mathcal{D}} \Big[ \exp\big( \tfrac{Q_\theta(s,a) - V_\phi(s)}{\beta} \big) \, \|a - f_\psi([s, \tilde{g}])\|_2^2 \Big], \tag{3}$$

where $\tau$ is the expectile level and $\beta$ is the temperature. At test time we condition $f_\psi$ on either $\tilde{g}_t$ or $s_{t+k}$ depending on the configuration.

### F.4 SUMMARY

- **Unified pipeline.** (1) Predict a keypoint-conditioned $k$-step target $s_{t+k}$; (2) realize control with either (A) a diffusion planner (with optional inverse dynamics) or (B) a GC-MLP trained with the IQL objective.

- **Pluggability.** Both backends implement the same interface $a_t = \text{LOWLEVEL}(s_t, \tilde{g}_t; \eta)$ and can be swapped without changing the high level.

- **Effect.** Bottleneck-guided subgoals provide reliable waypoints, so the low level only needs to execute short, simple transitions between keypoints.

## G HYPERPARAMETERS

We summarize the hyperparameters in Tab. 9. In all experiments we follow the ALLO configuration of Gómez et al. (2023) for the Laplacian encoder, except that we increase the embedding dimension from 11 to 21 to accommodate the more complex geometries. And we fix the cluster-crossing persistence threshold to $\tau = 20$ in all experiments. We observed that performance is insensitive to $\tau$ over a broad range, so we treat it as a fixed constant and do not tune it per environment.

## H LAPLACIAN REPRESENTATION

In this section, we present a series of visualizations of the Laplacian representation in various antmaze environments. The figures illustrate both the learned eigenvectors and the results of spectral clustering.

## I PROOFS AND TECHNICAL DETAILS FOR THEORIES

This appendix expands the statements in Section 3, provides self-contained proofs under standard assumptions, and aligns the notation with the main text. Throughout we work on a weighted,

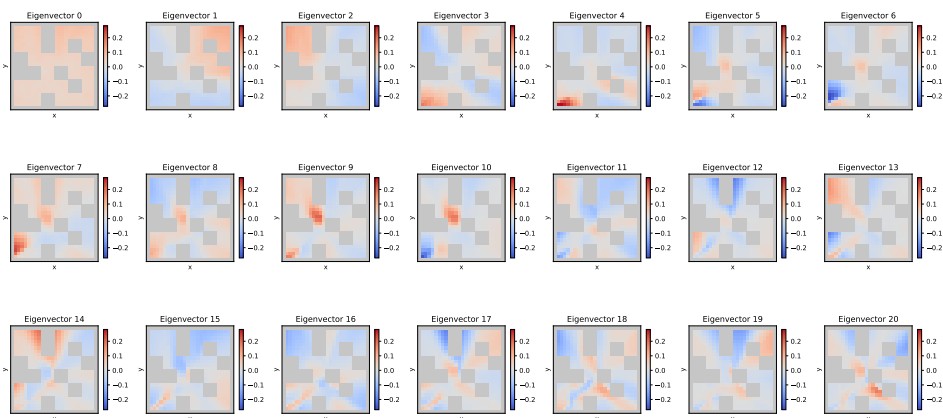

Figure 5: Learned eigenvectors for antmaze-medium-play

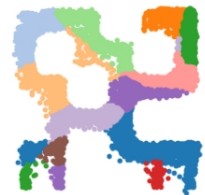

Figure 6: Spectral clustering results for antmaze-medium-play

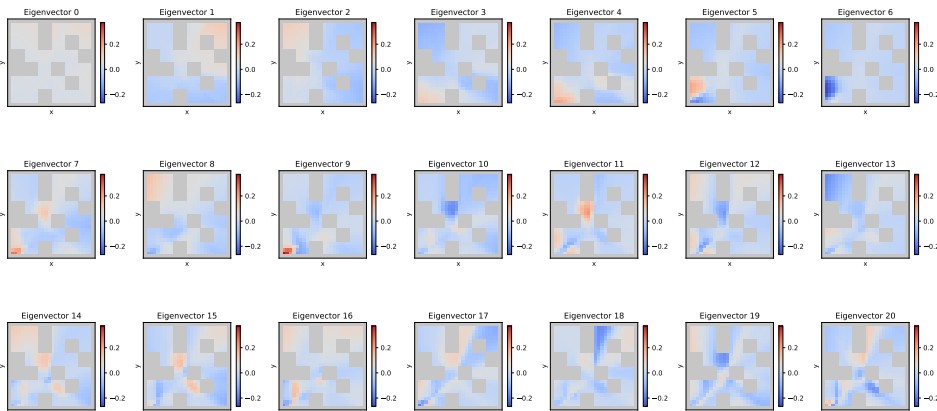

Figure 7: Learned eigenvectors for antmaze-medium-diverse

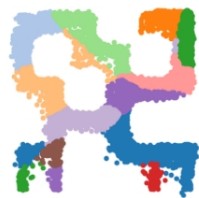

Figure 8: Spectral clustering results for antmaze-medium-diverse

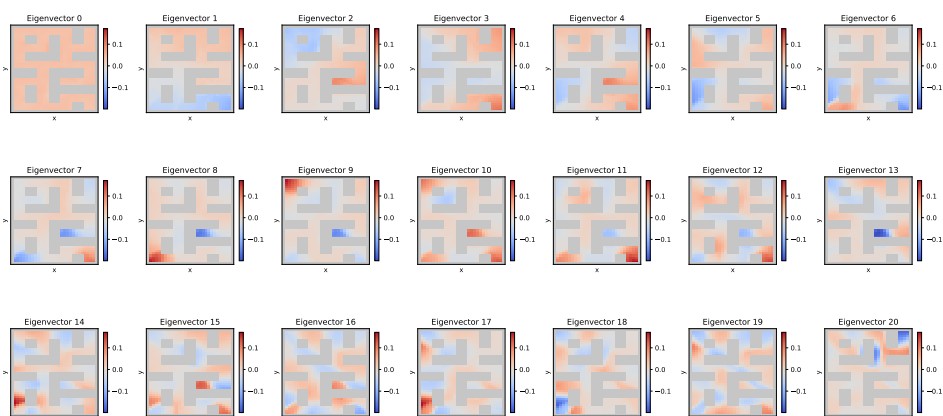

Figure 9: Learned eigenvectors for antmaze-large-play

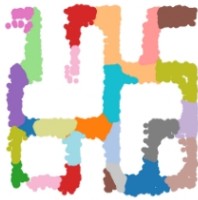

Figure 10: Spectral clustering results for antmaze-large-play

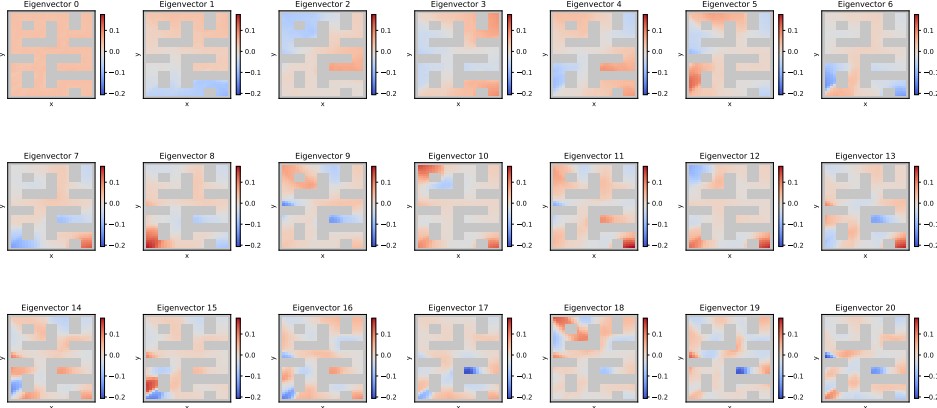

Figure 11: Learned eigenvectors for antmaze-large-diverse

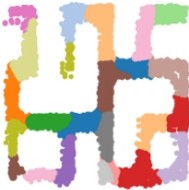

Figure 12: Spectral clustering results for antmaze-large-diverse

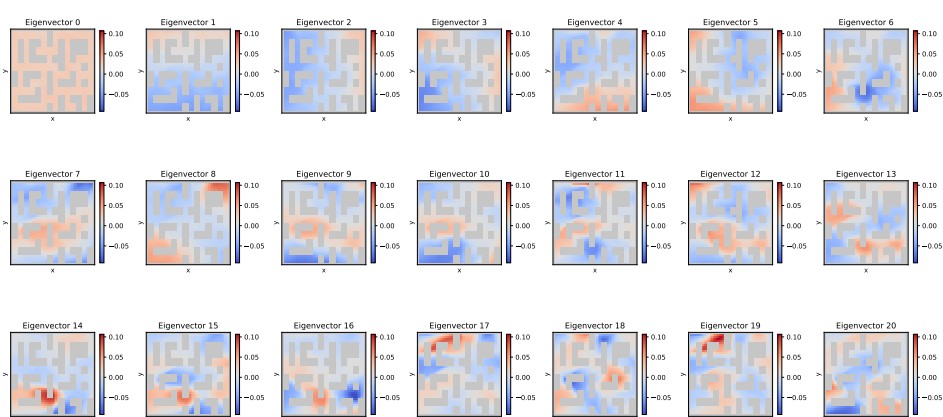

Figure 13: Learned eigenvectors for antmaze-ultra-play

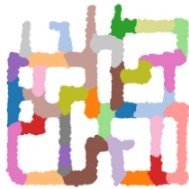

Figure 14: Spectral clustering results for antmaze-ultra-play

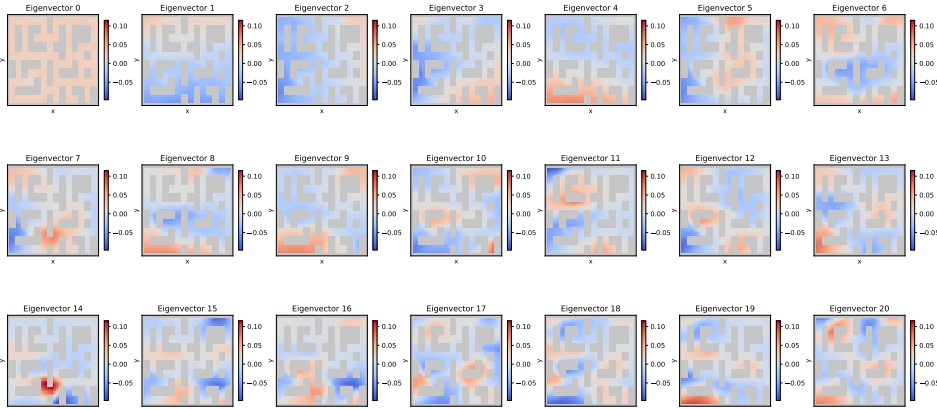

Figure 15: Learned eigenvector for antmaze-ultra-diverse

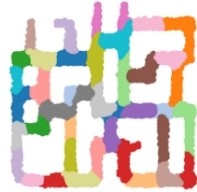

Figure 16: Spectral clustering results for antmaze-ultra-diverse

undirected state-transition graph $= (V, W)$ built from offline data, with degree $D = \operatorname{diag}(W\mathbf{1})$, random-walk kernel $P = D^{-1}W$, and random-walk Laplacian $L = I - P$. The (unique) stationary distribution is $\mu^\top P = \mu^\top$, $\sum_{v \in V} \mu(v) = 1$. Eigenvalues of $L$ satisfy $0 = \lambda_1 \leq \lambda_2 \leq \cdots$, and the $k$-way eigengap is $\gamma = \lambda_{k+1} - \lambda_k > 0$. For measurable $S, T \subseteq V$ define the inter-set flow

$$Q(S, T) = \sum_{u \in S, v \in T} \mu(u)P(u, v),$$

the (outer) conductance $\Phi(S) = Q(S, S^{\mathrm{c}})/\mu(S)$, and, for a region $R$, the internal conductance of the *reflected* chain $P_R$ (with stationary law $\mu_R$),

$$\Phi_{\mathrm{in}}(R) = \inf_{\emptyset \neq A \subseteq R, \, \mu_R(A) \leq \frac{1}{2}} \frac{Q_R(A, R \setminus A)}{\mu_R(A)}.$$

For $A \subseteq V$, write $\partial A = \{v \in V : P(v, A) > 0 \text{ and } P(v, A^{\mathrm{c}}) > 0\}$ and $\overline{A} = A \cup \partial A$. For any $B \subseteq V$ and $\varepsilon > 0$, $\mathcal{N}_\varepsilon(B)$ denotes the $\varepsilon$-neighborhood in the shortest-path (graph) metric. We use the following metastability condition (inner-strong / outer-weak):

$$\exists \, \mathcal{R}^\star = \{R_1^\star, \ldots, R_k^\star\} \text{ with } \mu(R_i^\star) \in [\eta, 1-\eta], \quad \Phi_{\mathrm{in}}(R_i^\star) \geq \alpha, \quad \Phi(R_i^\star \to R_j^\star) \leq \beta \ll \alpha \, (i \neq j). \tag{4}$$

Let $\widehat{L}$ be the Laplacian estimated from offline data and $\delta = \|\widehat{L} - L\|$ its operator-norm deviation.

**Hitting times and mixing.** For $A \subseteq V$, let $\tau_A = \inf\{t \geq 0 : X_t \in A\}$ be the hitting time, and define $T(x \to A) = \mathbb{E}_x[\tau_A]$. For a region $R$, let $t_{\mathrm{mix}}(R)$ be the least $t$ such that $\max_{x \in R} \|P_R^t(x, \cdot) - \mu_R\|_{\mathrm{TV}} \leq 1/4$; we write $t_{\mathrm{mix}}$ for $t_{\mathrm{mix}}(R_{\mathrm{cur}}^\star)$ when context is clear.

A. BOTTLENECK-GUIDED SUBGOAL OPTIMALITY (FULL VERSION OF THM. **??**)

**Theorem 3** (Bottleneck-first optimality). *Fix a start $s \in R_{\mathrm{cur}}^\star$ and a goal set $G \subseteq V \setminus R_{\mathrm{cur}}^\star$. Consider the one-step high-level objective*

$$\mathcal{J}(g) := T(s \to g) + T(g \to G), \qquad g \in V.$$

*Let $^\star$ denote a* next *mandatory cross-section for any $s \to G$ path (e.g., an $s$–$G$ minimum-capacity cut intersected with $\partial R_{\mathrm{cur}}^\star$), and let $\xi \sim \mathrm{FirstHit}(s, ^\star)$ be the first-hit distribution on $^\star$. Then there exists $g^\star \in \overline{^\star}$ such that*

$$\inf_{g \in V} \mathcal{J}(g) = T(s \to {}^\star) + \mathbb{E}_\xi[T(\xi \to G)] \pm C \cdot t_{\mathrm{mix}},$$

*where $C > 0$ is an absolute constant depending only on the chosen total-variation threshold in the definition of $t_{\mathrm{mix}}$.*

*Proof sketch.* (*Decomposition at the bottleneck*) By the strong Markov property at $\tau_\star$,

$$T(s \to g) = T(s \to {}^\star) + \mathbb{E}_\xi\big[T(\xi \to g)\big],$$

hence

$$\mathcal{J}(g) = T(s \to {}^\star) + \mathbb{E}_\xi\big[T(\xi \to g) + T(g \to G)\big].$$

(*Lower bound*) By the triangle inequality for hitting times, $T(\xi \to g) + T(g \to G) \geq T(\xi \to G)$ for any $g$, yielding

$$\mathcal{J}(g) \geq T(s \to {}^\star) + \mathbb{E}_\xi[T(\xi \to G)].$$

(*Achievability up to mixing*) Pick $g \in \overline{^\star}$. Inside $R_{\mathrm{cur}}^\star$, the reflected chain mixes to $\mu_{R_{\mathrm{cur}}^\star}$ in $t_{\mathrm{mix}}$ steps, so the distance from the first-hit $\xi$ to $g$ is controlled by $O(t_{\mathrm{mix}})$; likewise $T(g \to G) = \mathbb{E}_\xi[T(\xi \to G)] \pm O(t_{\mathrm{mix}})$. Combining with the decomposition gives the claim. $\square$

**Design implication (restated).** Placing the *next bottleneck* as the one-step subgoal is near-optimal up to an $O(t_{\mathrm{mix}})$ gap whenever movement inside a region is fast compared with crossing the bottleneck.

## B. Spectral clustering coverage of bottlenecks (full version of Thm. ??)

Let $U \in \mathbb{R}^{|V| \times k}$ collect the first $k$ nontrivial eigenvectors of $L$, and $Z$ be its row-normalization (each row scaled to unit $\ell_2$ norm). Likewise obtain $\widehat{U}, \widehat{Z}$ from $\widehat{L}$. Running $k$-means on the rows of $\widehat{Z}$ returns a partition $\widehat{\mathcal{R}} = \{\widehat{R}_1, \ldots, \widehat{R}_k\}$. Define the *misclustered volume* (up to permutation $\pi$) and the $\varepsilon$-thick *bottleneck overlap*:

$$\text{MisVol} = \min_{\pi \in S_k} \sum_{i=1}^{k} \mu(\widehat{R}_{\pi(i)} \triangle R_i^\star), \qquad \text{Overlap}_\varepsilon = 1 - \frac{\mu(\mathcal{N}_\varepsilon(\partial\widehat{\mathcal{R}}) \triangle \mathcal{N}_\varepsilon(\partial\mathcal{R}^\star))}{\mu(V)}.$$

**Theorem 4** (High-overlap bottleneck recovery). *Under metastability equation 4 with eigengap $\gamma = \lambda_{k+1} - \lambda_k > 0$ and empirical deviation $\delta = \|\widehat{L} - L\|$, there exist absolute constants $C_1, C_2, C_3 > 0$ such that*

$$\text{MisVol} \leq C_1 \frac{\beta}{\alpha} + C_2 \frac{\delta}{\gamma}, \qquad \text{Overlap}_\varepsilon \geq 1 - C_3 \text{MisVol} - \mu(\mathcal{N}_\varepsilon(\partial\mathcal{R}^\star)). \quad (5)$$

*Consequently, when $\beta/\alpha$ and $\delta/\gamma$ are small and the true bottleneck tube vanishes as $\varepsilon \downarrow 0$, the spectral partition achieves near-unity overlap with the true low-conductance bottlenecks.*

*Proof sketch. (i) Population embedding is region-constant up to $O(\beta/\alpha)$.* Write $L = L_0 + E$ with $L_0 = \text{blkdiag}(L_{R_1^\star}, \ldots, L_{R_k^\star})$ and $\|E\| \lesssim \beta$; each block has spectral gap $\lambda_2(L_{R_i^\star}) \gtrsim \alpha$. By Davis–Kahan/Weyl, the span of the first $k$ eigenvectors of $L$ deviates by $O(\beta/\alpha)$ from the ideal piecewise-constant subspace that is indicator-like on $\{R_i^\star\}$. Row-normalization maps the $k$ regions near the vertices of a regular simplex on $\mathbb{S}^{k-1}$, with separation bounded below by a constant depending on $(k, \eta)$.

*(ii) Empirical subspace stability is $O(\delta/\gamma)$.* With $\Delta = \widehat{L} - L$, Davis–Kahan yields $\|\sin\Theta(\widehat{U}, U)\| \leq C \delta/\gamma$. Thus each empirical row (of $\widehat{Z}$) lies within $\varepsilon_\star = C(\beta/\alpha + \delta/\gamma)$ of its ideal center on the unit sphere.

*(iii) $k$-means stability implies a misvolume bound.* Standard perturbation arguments for spherical $k$-means convert $\varepsilon_\star$ and center separation to $\text{MisVol} \leq C'\varepsilon_\star$ (up to constants depending on $(k, \eta)$), establishing the first inequality in equation 5.

*(iv) From misclustered volume to boundary overlap.* Misclustered points concentrate in a thin tube around the true inter-region boundaries; thickening by $\varepsilon$ absorbs local ambiguities and yields the overlap lower bound with a linear penalty in MisVol. $\qquad\square$

**Design implication (restated).** Learn a Laplacian embedding and cluster it. When within-region mixing is strong and cross-region transitions are rare (small $\beta/\alpha$), and the learned Laplacian is accurate relative to its eigengap (small $\delta/\gamma$), spectral clustering recovers bottlenecks with small error.

## C. Additional remarks and constants

**Choice of Laplacian.** All results extend to the symmetric normalized Laplacian $L_{\text{sym}} = I - D^{-1/2}WD^{-1/2}$ with the usual row/length normalizations; constants change by absolute factors.

**Estimating $\delta$.** In practice, $\delta$ is reduced by symmetrization, lazy random walks, density-regularized graphs, and sufficient offline coverage.

**Multiple comparable bottlenecks.** If several bottlenecks are comparable, $\lambda_2, \ldots, \lambda_k$ may be clustered; Theorem 4 still provides high overlap with their union. Our high-level planner then selects the next bottleneck along the cheapest $s \rightarrow G$ route (cf. Theorem 3).

**Mixing constant in Theorem 3.** The $O(t_{\text{mix}})$ term is with respect to the total-variation threshold $1/4$; other constants follow by standard monotonicity of total-variation mixing times.

*Summary.* The next bottleneck is the near-optimal one-step subgoal up to a small, interpretable mixing-time gap; and spectral clustering on a learned Laplacian recovers those bottlenecks with error controlled by the inner/outer conductance contrast and the Laplacian estimation error.

## J USE OF LLMS.

We used large language models solely for language polishing (grammar, wording, and clarity). They were *not* used to design experiments, generate or analyze data, write code, or substantively shape results or claims. All LLM-assisted edits were reviewed and verified by the authors.

