# OpenReview forum: "Don't Guess the Future, Find the Bottleneck: Spectral Subgoals for Offline Goal-Conditioned RL"
_ICLR.cc/2026/Conference — ICLR 2026 Conference Desk Rejected Submission_

### Official Review · Reviewer_3mmM · 2025-10-27

**Soundness:** 2
**Presentation:** 3
**Contribution:** 2
**Rating:** 4
**Confidence:** 4

**Summary:**

This paper proposes a new approach to hierarchical OGCRL. It first learns a representation of the state space and then applies K-means clustering to partition the space based on the learned representations. A graph is subsequently constructed to identify key points that must be traversed to reach the goal (although it is not entirely clear how these key points are selected as subgoals). Finally, the method trains a low-level policy to sequentially reach these key points, ultimately achieving the desired goal. The proposed approach demonstrates strong performance in long-horizon navigation tasks.

As suggested by the AC, I have added a list of the key claims made in the paper and briefly assessed their validity:

1. The optimal one-step subgoal is the next bottleneck (claimed in line 65). I would suggest being cautious with the use of the word "optimal". I can show a counterexample: in AntMaze, the ant can reach the goal through the inner circle of the branch rather than the bottleneck in the middle of the branch.

2. The proposed method shows consistent bottleneck recovery. This seems reasonable as shown in Figure 3, but it would be more convincing to include results from different environments rather than only navigation tasks.

3. The proposed method brings performance gains. I would agree, as there are significant improvements over the baselines (Table 2).

4. The method generalises well to different but similar environments. This is supported by the high-level and low-level transfer experiments, which are both interesting and convincing.

5. The claim on the impact of the number of clusters $K$ is not thoroughly verified, since the experiments are conducted in only one environment.

**Strengths:**

Overall, this is a good paper with a clear presentation, a novel idea, and strong empirical performance.

**Weaknesses:**

Compared to other work in OGCRL, the proposed method appears somewhat complicated, although this is not necessarily a drawback. The authors should provide the hyperparameters used in their experiments to demonstrate the robustness or potential vulnerability of the proposed approach. Furthermore, additional ablation studies should be included in the paper, as suggested in Question 5.

**Questions:**

1. The sentence "don’t guess the future, find the bottleneck" is somewhat ambiguous. From my reading, I understand that the authors mean “we do not rely on remaining time or value guidance; instead, we propose to follow the bottleneck.” Therefore, the authors may consider revising the title and the statement in line 62 for greater clarity.

2. OGCRL typically employs a state-to-goal mapping, often denoted as $\phi$ in related literature. For example, the state $s$ in AntMaze includes the full dynamics of the ant (such as its location), while $\phi(s)$ represents a slice of $s$, capturing only the ant’s location. In this paper, BASS learns a Laplacian encoder $\phi_\theta$ and uses the resulting latent code for state clustering (via K-means, as mentioned in line 199). My question is whether this representation learning step is necessary. Could we instead directly use $\phi(s)$ for clustering? Based on my experience, this might already yield a similar structure to what is shown in Figure 1.

3. What does $D$ denote in line 213? The state coordinate dropping requires further clarification.

3. There is a duplicated sentence in lines 243–245. Within that sentence, the meaning of goal residual is unclear. It is also difficult to understand how the authors select the next KP. Do the authors plan the shortest sequence of KPs from $s_0$ to $s_{goal}$? If so, does this shortest sequence in the graph correspond to the shortest path from $s_0$ to $s_{goal}$?

4. The proposed method is relatively more complicated than other OGCRL approaches; therefore, it is necessary to conduct more thorough ablation studies. For example, they could examine the impact of the representation learning step discussed in Question 2, the hyperparameter $\tau$ in line 202, and the number of K-means clusters. Although the authors analysed the effect of the number of clusters $K$, conducting this analysis in only one environment is insufficient to fully support their claim.

---

> ### Author Response · Authors · 2025-11-23
> **Response to Reviewer 3mmM(Part 1/3)**
>
> ---
>
> ### W1. On “the optimal one-step subgoal is the next bottleneck”
>
> Our use of the word “optimal” is meant in a theoretical sense, under an idealized metastable setting: we assume the state space is partitioned into regions that are internally well-mixed and connected to each other only through a small set of bottleneck states. In this regime, the one-step subgoal should be placed on the bottleneck states on the interface between two regions, rather than deep inside either metastable region, because those interface states minimize the expected hitting time to the next region in the induced Markov chain.
>
> We have already revised the abstract to clarify that our statements hold under a standard metastable decomposition of the state space. In the theoretical takeaway section, even in the initial submission we had already acknowledged that, due to various sources of approximation error, in practice choosing subgoals at the discovered bottlenecks yields **near-optimal** plans.
>
> In practice, when we construct subgoals, we do not try to pick a single infinitesimal “point” on the boundary, but instead take the mean of the empirical bottleneck states in that interface. This mean typically lies closer to the middle of the corridor rather than hugging the inner wall. In AntMaze specifically, while an inner-ring path may be shorter in pure geometric distance, it has a high probability of colliding with walls or corners, which slows down the ant and often causes it to fall, thereby increasing the expected number of steps to reach the goal. In contrast, the “middle corridor” bottleneck that BASS discovers is the more robust bottleneck under the actual dynamics and offline data and in practice leads to a shorter expected crossing time.
>
> ---
>
> ### W2. On showing bottleneck recovery beyond navigation tasks
>
> Thank you for the suggestion. We would like to clarify that BASS has already been evaluated beyond low-dimensional navigation tasks.
>
> **FrankaKitchen (robotic manipulation).**
> On the widely used offline GCRL benchmarks kitchen-mixed and kitchen-partial, BASS achieves state-of-the-art success rates and clearly outperforms HIQL. These tasks involve high-dimensional robot and object configurations, indicating that our bottleneck discovery is not restricted to 2D maze layouts.
>
> **Visual AntMaze (image observations).**
> In the rebuttal, we additionally evaluate BASS on visual-antmaze-large-navigate and visual-antmaze-large-stitch, where the agent receives raw image observations that are first encoded into a latent space before we build the Laplacian graph. BASS again significantly improves over HIQL:
>
> | Dataset                       | HIQL   | BASS (Ours) |
> | ----------------------------- | ------ | ----------- |
> | kitchen-partial-v0            | 65.0   | 83.3 ± 4.9  |
> | kitchen-mixed-v0              | 67.7   | 86.0 ± 2.8  |
> | visual-antmaze-large-navigate-v0 | 53 ± 9 | 78.7 ± 2.3  |
> | visual-antmaze-large-stitch-v0   | 28 ± 2 | 68.0 ± 2.0  |
>
> ---
>
> ### Q1. revising the title and the statement in line 62
>
> Thank you for this suggestion. We agree that the current slogan-style phrasing is somewhat ambiguous and does not clearly reflect our technical focus. In the revised version, we will:
>
> * Change the title to
>   “Bottleneck-Guided Spectral Subgoals for Offline Goal-Conditioned RL”.
> * Rewrite the sentence around line 62 to a more precise statement:
>   “We argue that high-level subgoals in offline GCRL should not be chosen via value estimates or time heuristics, but instead be derived from bottleneck states revealed by the Laplacian spectral structure of the offline data.”
>
> We believe this revision better matches your interpretation (“not relying on remaining time or value guidance, but following bottlenecks”) while making the message clearer and more aligned with the technical content of the paper.
>
> ---

---

> > ### Author Response · Authors · 2025-11-23
> > **Response to Reviewer 3mmM(Part 2/3)**
> >
> > ### Q2. Is Laplacian representation learning necessary, or could we just cluster φ(s)?
> >
> > This idea is closely related to recent work such as QPHIL (“Navigation with QPHIL: Quantizing Planner for Hierarchical Implicit Q-Learning”, arXiv:2411.07760). QPHIL learns a VQ-VAE quantizer on the state space and treats each codebook vector as a landmark, clustering states by Euclidean distance in that space and then planning over those discrete tokens.
> >
> > Our view is that this line of work is precisely what “clustering φ(s)” looks like in practice, and it highlights the main limitation of purely geometric clustering: the clusters primarily capture local geometry but do not explicitly encode reachability. For example, in QPHIL’s own visualizations on AntMaze (their Fig. 5), several tokens span both sides of a wall, grouping states that are close in Euclidean coordinates but require long detours to reach each other in the true transition graph. Such landmarks may form reasonable geometric zones, but they are clearly not the most informative keypoints for guiding the agent’s movement.
> >
> > In contrast, BASS first learns a Laplacian representation and then performs spectral clustering in that space, so that distances and cluster boundaries are explicitly shaped by the random-walk connectivity induced by the offline dataset. We then extract keypoints at intersections of trajectories that cross clusters and build a directed reachability graph over these bottleneck keypoints. This procedure is designed to expose metastable regions and the low-conductance interfaces between them, rather than just grouping geometrically nearby states.
> >
> > Empirically, we compared QPHIL’s results to ours on the shared environments:
> >
> > | Env                       | QPHIL  | BASS（ours）  |
> > | ------------------------- | ------ | ----------- |
> > | antmaze-medium-play-v2    | 91 ± 2 | 98.0 ± 0.0  |
> > | antmaze-medium-diverse-v2 | 92 ± 4 | 96.7 ± 0.9  |
> > | antmaze-large-play-v2     | 80 ± 3 | 96.0 ± 1.6  |
> > | antmaze-large-diverse-v2  | 82 ± 6 | 98.7 ± 1.9  |
> > | antmaze-ultra-play-v0     | 62 ± 4 | 97.3 ± 0.9 |
> > | antmaze-ultra-diverse-v0  | 62 ± 7 | 88.0 ± 1.6 |
> >
> > Across medium, large, and ultra mazes, BASS consistently improves success by about +4–7 points (medium), +16–17 points (large), and 25+ points (ultra). This systematic gap suggests that, while clustering φ(s) (or a VQ-VAE embedding of it) can indeed produce a reasonable partition of the space, it is not sufficient to reliably discover the true bottlenecks that dominate long-horizon success. The Laplacian representation learning step is therefore not just cosmetic: it is what aligns the clustering with hitting-time bottlenecks, which in turn yields more effective subgoals for offline GCRL.
> >
> > ---
> >
> > ### Q3. Clarification on (D) and state coordinate dropping
> >
> > Thank you for pointing out this ambiguity. In Sec. 4.1, the notation (S \subset \mathbb{R}^D) uses (D) to denote the dimensionality of the filtered state vector that we use to construct the Laplacian for spectral clustering. Before building the Laplacian, we apply a simple and fully automatic filtering step: for each coordinate, we estimate how fast it varies along trajectories and retain only slowly varying, temporally smooth dimensions (e.g., positions, object poses), while discarding highly oscillatory ones (e.g., joint velocities), following the low-frequency intuition of Laplacian RL. The quantity (D) is the dimension of this filtered vector.
> >
> > The “state coordinate dropping’’ in Sec. 4.2 is then used to make each keypoint applicable to unseen goal compositions, especially in manipulation tasks. For a given bottleneck crossing, we record only the coordinates that consistently change when the KP is activated (e.g., the pose of a moved object or the angle of a rotated switch), while ignoring coordinates that remain unaffected.
> >
> > Intuitively, this allows a single KP to be reused under many different global configurations: for example, a KP corresponding to “move the kettle onto the stove’’ can fire regardless of the states of other objects or cabinet doors, as long as the coordinates related to the kettle and stove match. This is precisely what enables BASS to handle goal combinations that never appear explicitly in the offline dataset.
> >
> > (As clarified elsewhere in the rebuttal, this “filtering” and “dropping’’ affects only the high-level semantic encoding of KPs and the planner on the KP graph; the actual subgoal passed to the low-level controller is always the full underlying state.)
> >
> > ---

---

> > > ### Author Response · Authors · 2025-11-23
> > > **Response to Reviewer 3mmM(Part 3/3)**
> > >
> > > ### Q4. Goal residual and how we select the next KP
> > >
> > > After revisiting the text, we found that introducing this term is unnecessary for understanding the planner, so in the revision we will replace
> > > “a shortest sequence that eliminates the remaining goal residual’’
> > > with the simpler description:
> > > “a shortest sequence that drives the agent from the current state to the goal.’’
> > >
> > > Here, “a shortest sequence’’ means the sequence of KPs that reaches the goal with the fewest KPs. It is noted that we focus on minimizing the number of difficult bottleneck transitions, rather than the geometric length of the path in the original state space.
> > >
> > > ---
> > >
> > > ### Q5. Ablations on important hyper parameters
> > >
> > > **Representation-learning step.**
> > > The additional complexity in BASS mainly comes from the Laplacian representation + spectral clustering block. For the Laplacian encoder, we follow the hyperparameter settings by Gómez et al. (2023) almost exactly; the only substantive change is the embedding dimension: because our environments aremore complex than the gridworld-style tasks used in Gómez et al. (2023), we increase the embedding dimension (d) from 11 to 21, i.e., 1 zero-eigenvector + 20 low-frequency eigenvectors, to capture more low-frequency structure. We will clarify this design choice in the paper.
> > >
> > > **τ in clustering.**
> > > The parameter (\tau = 20) mentioned in line 202 is an early, fixed constant used to require that a cluster crossing persists for at least (\tau) steps before we record a candidate keypoint. We did not tune this value. We will make this clearer and de-emphasize (\tau) in the main text, since it is not a sensitive hyperparameter.
> > >
> > > **Number of clusters K.**
> > > The K-means cluster count (K) is the most influential hyperparameter in the spectral clustering stage. In the revision, we extend this analysis to four environments (antmaze-large-play, antmaze-large-diverse, pointmaze-large-stitch, humanoidmaze-large-stitch) and sweep a much wider range of (K).
> > >
> > > Empirically, the extended results show a very consistent pattern:
> > > For each environment, we sweep (K) over a much wider range (e.g., 10–50 in large and 30–60 in giant).
> > >
> > > For large environments:
> > >
> > > | Env / K                | 10         | 15          | 20          | 22          | 24         | 26          | 28          | 30         | 32          | 34             | 36          | 38         | 40          | 42          | 44          | 46          | 48          | 50         |
> > > | ---------------------- | ---------- | ----------- | ----------- | ----------- | ---------- | ----------- | ----------- | ---------- | ----------- | -------------- | ----------- | ---------- | ----------- | ----------- | ----------- | ----------- | ----------- | ---------- |
> > > | antmaze-large-play     | 32.0 ± 4.0 | 15.3 ± 7.0  | 96.0 ± 1.6  | 93.3 ± 3.1  | 94.7 ± 3.1 | 98.0 ± 2.0  | 91.3 ± 3.1  | 91.3 ± 3.1 | 95.3 ± 1.9  | **98.0 ± 0.0** | 92.0 ± 1.6  | 96.7 ± 0.9 | 90.7 ± 3.4  | 87.3 ± 5.0  | 94.0 ± 3.3  | 89.3 ± 8.1  | 90.0 ± 3.3  | 79.3 ± 4.1 |
> > > | pointmaze-large-stitch | 98.0 ± 1.6 | 100.0 ± 0.0 | 100.0 ± 0.0 | 100.0 ± 0.0 | 99.3 ± 0.9 | 100.0 ± 0.0 | 100.0 ± 0.0 | 98.0 ± 3.5 | 100.0 ± 0.0 | 100.0 ± 0.0    | 100.0 ± 0.0 | 96.0 ± 2.0 | 100.0 ± 0.0 | 100.0 ± 0.0 | 100.0 ± 0.0 | 100.0 ± 0.0 | 100.0 ± 0.0 | 96.7 ± 1.2 |
> > >
> > > For giant environments:
> > >
> > > | Env / K                | 30             | 32         | 34         | 36         | 38         | 40         | 42         | 44         | 46         | 48         | 50             | 52         | 54         | 56         | 58         |
> > > | ---------------------- | -------------- | ---------- | ---------- | ---------- | ---------- | ---------- | ---------- | ---------- | ---------- | ---------- | -------------- | ---------- | ---------- | ---------- | ---------- |
> > > | antmaze-giant-stitch   | 0.0 ± 0.0      | 13.3 ± 2.3 | 20.0 ± 7.0 | 11.3 ± 3.1 | 40.0 ± 8.0 | 53.3 ± 3.1 | 40.7 ± 4.2 | 60.0 ± 9.2 | 62.0 ± 6.0 | 68.0 ± 3.5 | **71.3 ± 7.0** | 63.3 ± 2.3 | 66.7 ± 2.3 | 66.7 ± 5.0 | 61.3 ± 3.1 |
> > > | pointmaze-giant-stitch | **92.0 ± 2.0** | 84.7 ± 1.2 | 80.7 ± 3.1 | 86.7 ± 3.1 | 90.0 ± 5.3 | 81.3 ± 1.2 | 78.7 ± 4.2 | 84.7 ± 5.1 | 80.0 ± 3.5 | 88.7 ± 2.3 | 85.3 ± 3.1     | 83.3 ± 6.1 | 85.3 ± 6.1 | 88.7 ± 6.1 | 84.7 ± 3.1 |
> > >
> > > The new results show a consistent pattern across environments:
> > >
> > > * (i) Very small (K) leads to overly coarse partitions, under-detecting bottlenecks and hurting performance;
> > > * (ii) There is a broad plateau of (K)-values where performance is stable and often matches or even exceeds the numbers reported in the main submission;
> > > * (iii) In a few environments, very large (K) can slightly reduce performance due to unnecessary path complexity, which is consistent with your concern about “stuck” states in corners, but the degradation remains moderate. Notably, on antmaze-giant-stitch, increasing (K) actually uncovers a substantially better operating point (around 61–71% success) than what we originally reported in the initial submission.

---

> > > > ### Author Response · Authors · 2025-11-27
> > > > **Thank you for your updated evaluation**
> > > >
> > > > We sincerely thank you for your careful review and for reconsidering your evaluation after our rebuttal. We are very happy that our clarifications resolved your questions, and we greatly appreciate your recognition of the contributions and potential impact of our work.

---

### Official Review · Reviewer_Ne7X · 2025-10-28

**Soundness:** 2
**Presentation:** 3
**Contribution:** 2
**Rating:** 4
**Confidence:** 4

**Summary:**

The paper proposes an offline goal-conditioned reinforcement learning (OGCRL) method that learns to expose bottlenecks as subgoals and discover trajectories to cross these subgoals. Specifically, key points (KPs) on the cluster boundaries are revealed as bottlenecks, and a KP graph is constructed for further route planning. Experimental results on D4RL and OGBench show improved performance of the proposed method compared with representative offline methods.

**Strengths:**

1. The proposed framework is clear and easy to follow.
2. Leveraging state transitions to select boundaries is interesting, as the key points could indicate hard-to-cross regions.
3. Experimental results show that the proposed method achieves an improved success rate compared with previous works.

**Weaknesses:**

1. I have several concerns regarding the selection of key points (KPs):
- First of all, KP selection is highly dependent on the clustering performance. As discussed in Section 5.4, reducing the number of clusters results in coarse discovered boundaries. In addition, increasing the number of clusters might introduce route complexity during planning. Although Section 5.4 includes an ablation study, the range of cluster numbers evaluated is, in my opinion, insufficient. Since the peak success rate is achieved when K=26, only K=28 is evaluated as an “extra cluster,” and the drop in success rate (6.7%) is non-trivial. I would recommend evaluating more cluster numbers. Furthermore, as route complexity may increase with more KPs, adding trajectory steps could strengthen the experiment.
- Secondly, in practice, the offline dataset is often noisy, where the agent may reach unexpected states and get stuck (e.g., corners in AntMaze). In such cases, complex yet unnecessary KPs may appear, causing additional complexity during planning.

2. The idea of exposing cluster boundaries as subgoals is interesting, but the similar idea has been studied in this paper (https://arxiv.org/pdf/2411.01396).

3. Some questions regarding the claim that “KPs are optimal subgoals”:
- To sufficiently support this claim, I would recommend adding trajectory steps in the experimental results for comparison with the baselines.
- In addition to Section 5.4, a comparison of trajectory steps with high-quality demonstrations or expert-annotated routes could further strengthen the evidence.

4. Minor Issues:
- The visualization of KPs is interesting, but the visualization of trajectories seem not to be shown in Figure 3.
- KP discovery in Section 4.1 is an essential and complex step of the proposed method. Including pseudocode could strengthen the readability.

**Questions:**

Please answer the questions in weakness.

---

> ### Author Response · Authors · 2025-11-23
> **Response to Reviewer Ne7X(Part 1/3)**
>
> ### Q1. Sensitivity of KP selection to the number of clusters, and potential spurious KPs from noisy offline data.
>
> (a) Sensitivity to the number of clusters (K).
> In the original submission, the ablation on (K) was indeed limited to a single environment. Following your suggestion, we have significantly extended this study:
>
> * We now report (K)-ablations on four representative benchmarks: antmaze-large-play, pointmaze-large-stitch, antmaze-giant-stitch, and pointmaze-giant-stitch.
> * For each environment, we sweep (K) over a much wider range (e.g., 10–50 in large env and  30–60 in giant env).
>
> For large environments:
>
> | Env / K                | 10         | 15          | 20          | 22          | 24         | 26          | 28          | 30         | 32          | 34             | 36          | 38         | 40          | 42          | 44          | 46          | 48          | 50         |
> | ---------------------- | ---------- | ----------- | ----------- | ----------- | ---------- | ----------- | ----------- | ---------- | ----------- | -------------- | ----------- | ---------- | ----------- | ----------- | ----------- | ----------- | ----------- | ---------- |
> | antmaze-large-play     | 32.0 ± 4.0 | 15.3 ± 7.0  | 96.0 ± 1.6  | 93.3 ± 3.1  | 94.7 ± 3.1 | 98.0 ± 2.0  | 91.3 ± 3.1  | 91.3 ± 1.2 | 95.3 ± 1.9  | **98.0 ± 0.0** | 92.0 ± 1.6  | 96.7 ± 0.9 | 90.7 ± 3.4  | 87.3 ± 5.0  | 94.0 ± 3.3  | 89.3 ± 8.1  | 90.0 ± 3.3  | 79.3 ± 4.1 |
> | pointmaze-large-stitch | 98.0 ± 1.6 | 100.0 ± 0.0 | 100.0 ± 0.0 | 100.0 ± 0.0 | 99.3 ± 0.9 | 100.0 ± 0.0 | 100.0 ± 0.0 | 98.0 ± 3.5 | 100.0 ± 0.0 | 100.0 ± 0.0    | 100.0 ± 0.0 | 96.0 ± 2.0 | 100.0 ± 0.0 | 100.0 ± 0.0 | 100.0 ± 0.0 | 100.0 ± 0.0 | 100.0 ± 0.0 | 96.7 ± 1.2 |
>
> For giant environments:
>
> | Env / K                | 30         | 32         | 34         | 36         | 38         | 40         | 42         | 44         | 46         | 48         | 50         | 52         | 54         | 56         | 58         |
> | ---------------------- | ---------- | ---------- | ---------- | ---------- | ---------- | ---------- | ---------- | ---------- | ---------- | ---------- | ---------- | ---------- | ---------- | ---------- | ---------- |
> | antmaze-giant-stitch   | 0.0 ± 0.0  | 13.3 ± 2.3 | 20.0 ± 7.0 | 11.3 ± 3.1 | 40.0 ± 8.0 | 53.3 ± 3.1 | 40.7 ± 4.2 | 60.0 ± 9.2 | 62.0 ± 6.0 | 68.0 ± 3.5 | **71.3 ± 7.0** | 63.3 ± 2.3 | 66.7 ± 2.3 | 66.7 ± 5.0 | 61.3 ± 3.1 |
> | pointmaze-giant-stitch | **92.0 ± 2.0** | 84.7 ± 1.2 | 80.7 ± 3.1 | 86.7 ± 3.1 | 90.0 ± 5.3 | 81.3 ± 1.2 | 78.7 ± 4.2 | 84.7 ± 5.1 | 80.0 ± 3.5 | 88.7 ± 2.3 | 85.3 ± 3.1 | 83.3 ± 6.1 | 85.3 ± 6.1 | 88.7 ± 6.1 | 84.7 ± 3.1 |
>
> The average trajectory steps for different values of K are:
>
> | K / Env | pointmaze-giant-stitch | antmaze-giant-stitch |
> | --------- | ------------------------- | ----------------------- |
> | 30        | **497.79**       | 1000.00          |
> | 32        | 515.02           | 979.45          |
> | 34        | 518.50           | 927.49         |
> | 36        | 509.70           | 972.08          |
> | 38        | 503.09           | 929.33         |
> | 40        | 517.41           | 858.93         |
> | 42        | 524.04           | 893.85         |
> | 44        | 513.80           | 860.83         |
> | 46        | 502.10           | 845.30         |
> | 48        | 503.00           | 810.18         |
> | 50        | 510.25           | **791.27**     |
> | 52        | 531.18           | 805.50         |
> | 54        | 516.12           | 802.30         |
> | 56        | 498.68           | 800.60         |
> | 58        | 508.03           | 810.87         |
>
> The new results show a consistent pattern across environments:
>
> - Very small (K) leads to overly coarse partitions, under-detecting bottlenecks and hurting performance;
> - There is a broad plateau of (K)-values where performance is stable and often matches or even exceeds the numbers reported in the main submission;
> - In a few environments, very large (K) can slightly reduce performance due to unnecessary path complexity, which is consistent with your concern about “stuck” states in corners, but the degradation remains moderate. Notably, on antmaze-giant-stitch, increasing (K) actually uncovers a substantially better operating point (around 61–71% success) than what we originally reported in the initial submission.
>
> These extended results support our claim that BASS is not overly sensitive to the exact choice of (K): extreme values (too small or too large) can hurt performance, but there is a robust region where the discovered KPs and the resulting plans are stable.
>
> ---

---

> ### Author Response · Authors · 2025-11-23
> **Response to Reviewer Ne7X(Part 2/3)**
>
> ### Q2: Relation to CE² and the use of cluster boundaries
> We thank the reviewer for pointing us to CE² ([https://arxiv.org/pdf/2411.01396](https://arxiv.org/pdf/2411.01396)), which also makes use of cluster boundaries.  We will add this work to our related work. While both approaches touch cluster boundaries, the setting, objective, use of these boundaries, and clustering method are quite different.
>
> * Setting and objective. CE² is an **online, unsupervised, goal-conditioned exploration algorithm**: it clusters states in a latent space under the current policy, and then samples “near-boundary but still reachable” states as exploratory goals to improve exploration efficiency. In contrast, our work targets offline goal-conditioned RL (OGCRL): **there is no online interaction, and our focus is on how to better leverage a fixed offline dataset to identify bottlenecks in the state space and perform long-horizon hierarchical planning**. In our case, subgoals are used by a high-level planner to route on a keypoint graph in order to solve “long horizon + sparse reward” planning problems, rather than to drive exploration. In this sense, the two works study complementary aspects of RL: online exploration versus offline exploitation.
> * Role and form of boundaries. In CE², the “boundary” notion is used to select frontier goals that are still relatively easy to reach, and these are used during training to encourage exploration. They **do not form an explicit keypoint / bottleneck graph structure**. In our method, we first learn a Laplacian representation and apply spectral clustering to expose metastable regions, then extract the intersection states of trajectories that cross clusters as keypoints, and finally build a directed reachability graph (G_{\mathrm{KP}}) over these bottleneck keypoints. The high-level policy plans entirely on this graph, and the low-level controller only executes short transitions between successive keypoints.
> * Clustering method. CE² clusters latent states directly in the representation space using a standard geometric notion of distance (e.g., Euclidean distance in the **latent embedding derived from a world model**), without explicitly encoding the random-walk connectivity structure of the MDP. In contrast, we learn a Laplacian representation and then perform spectral clustering on this **Laplacian embedding**, so the boundaries of these clusters are shaped by metastable regions and low-conductance cuts in the underlying transition graph.
>
> Put differently, CE² performs goal sampling on latent clusters to guide online exploration, whereas our method performs hierarchical planning on a bottleneck graph constructed from Laplacian spectral structure in an offline setting. The two approaches are therefore complementary in purpose and structure, despite both referring to cluster “edges” or boundaries.
>
> ---
>
> ### Q3. Trajectory length comparison for bottleneck subgoals
>
> To address the reviewer’s suggestion, we have added average evaluation trajectory lengths on OGBench in the following table:
>
> | Dataset                        | HIQL Steps | BASS (ours) Steps |
> | ------------------------------ | ---------- | ----------------- |
> | antmaze-giant-stitch-v0        | 997.50     | 864.17            |
> | antmaze-large-stitch-v0        | 640.87     | 547.28            |
> | pointmaze-large-stitch-v0      | 905.04     | 265.33            |
> | humanoidmaze-large-navigate-v0 | 1667.65    | 1652.34           |
> | humanoidmaze-large-stitch-v0   | 1808.40    | 1717.22           |
> | humanoidmaze-giant-navigate-v0 | 3900.32    | 3287.94           |
> | humanoidmaze-giant-stitch-v0   | 3978.45    | 3319.76           |
>
> Across all ogbench environments, BASS attains shorter trajectories than HIQL while also achieving higher success rates (see main OGBench results), indicating that routing via bottleneck keypoints yields more direct paths in the state space rather than merely “lucky” successes. These results suggest that bottleneck-aligned keypoints help the agent cross critical corridors and transitions more reliably, and are consistent with our theoretical claim that keypoints at bottlenecks serve as near-optimal one-step subgoals in terms of hitting-time efficiency.
>
> ---

---

> > ### Author Response · Authors · 2025-11-23
> > **Response to Reviewer Ne7X(Part 3/3)**
> >
> > ### Q4 & Q5. Trajectory length and comparison with expert routes
> > Thank you for this suggestion. In the revised version we add a trajectory visualization on antmaze-large-diverse, where we directly compare rollouts guided by BASS keypoints with rollouts guided by a hand-designed “oracle” keypoint sequence (at the center of each junction). Both use the same low-level controller; the only difference is whether the intermediate subgoals come from BASS or from human-specified waypoints. As shown in Figure 4 of the revision (near line 760), trajectories under BASS and under the oracle keypoints pass through essentially the same narrow corridors. Both methods achieve success rates above 97%, and their average episode lengths are **724.85 ± 99.83 (ours) and 721.04 ± 112.11 (oracle)**, indicating that our keypoints provide highly accurate coverage of the bottlenecks.
> >
> > ### Q6. Pseudocode of Section 4.1
> > Due to space limitations, we provide the pseudocode of this procedure in the appendix A of the revision (near line 649).
> >
> > ```text
> > Algorithm 1  Bottleneck keypoint discovery
> >
> > Input:   offline dataset D_off = {h_i}, number of clusters K,
> >          boundary persistence τ_b
> > Output:  keypoint set KPs
> >
> > 1:  // Laplacian representation and spectral clustering
> > 2:  Train a Laplacian encoder φ on states { s | (s, ·) ∈ D_off } and obtain
> > 3:      embeddings z = φ(s).
> > 4:  Run K-means with K clusters on { z }; let c(s) ∈ {1,…,K} be the cluster
> > 5:      label of state s.
> >
> > 6:  // Collect boundary samples between clusters
> > 7:  Initialize boundary buffer B ← ∅.
> > 8:  For each trajectory h = (s_0,…,s_T) in D_off do
> > 9:      For t = 0,…,T − τ_b − 1 do
> > 10:          if c(s_t) ≠ c(s_{t+1}) and
> > 11:               c(s_{t+1}) = … = c(s_{t+τ_b}) then
> > 12:              Append s_{t+1} to B          // candidate boundary state
> > 13:          end if
> > 14:      end for
> > 15:  end for
> >
> > 16: // Compress boundary samples into keypoints
> > 17: Initialize keypoint set KPs ← ∅.
> > 18: Group boundary samples in B into small neighborhoods of nearby states
> > 19:     and compute a representative center μ_ℓ for each group.
> > 20: For each center μ_ℓ do
> > 21:     Construct a keypoint KP_ℓ = (I_Δ, v_Δ) from μ_ℓ as described in Sec. 4.2.
> > 22:     Add KP_ℓ to KPs.
> > 23: end for
> >
> > 24: return KPs
> > ```

---

### Official Review · Reviewer_gT9f · 2025-10-29

**Soundness:** 3
**Presentation:** 3
**Contribution:** 2
**Rating:** 6
**Confidence:** 3

**Summary:**

This paper proposed a novel algorithm for generating a planning graph for the offline goal-conditioned reinforcement learning setting. The method works by discovering bottlenecks via a Laplacian Spectral Clustering Algorithm (specifically, ALLO). With this, they have a novel mechanism for identifying the optimal bottleneck to user as the next subgoal in a planning algorithm. The plan of subgoals is that leveraged with a diffusion planner or an MLP for generating actions using subgoals generated by a planner. The key contribution is a mechanism for generating nodes in a graph based on "keypoints" derived from ALLO, along with theoretical results for why these are the "best" nodes for planning.

**Strengths:**

The theoretical foundation and the algorithm are very clearly explained.

The writing is clear as well.

The results are strong with the baselines they consider. They have many baselines, which is good. They use 3 environments: AntMaze, FrankaKitchen, and Maze2D, which is good.  They show generalization results across environments, which is good. Presence of ablations are also good.

**Weaknesses:**

It’s unclear what the specific contribution of this paper is regarding the use of Laplacian graph clusters for subgoal identification compared to prior work. The authors claim that “none of the existing methods have been used to significantly enhance goal-conditioned decision-making,” but that doesn’t seem accurate. There are numerous papers that leverage Laplacian-based representations for goal-conditioned reinforcement learning (RL), where Laplacian bottlenecks are explicitly used to discover meaningful subgoals. Such representations have already been shown to benefit RL. Is it that most of those focused on the benefits for exploration, whereas this paper focuses on offline RL?

If the authors are arguing that their setting is distinct because it involves transferring from an offline dataset to multiple goals simultaneously, that distinction needs to be clarified. It’s also not clear from the evaluation description how generalization is being tested for each offline dataset---are we evaluating on 1 goal, 4 goals, 10 goals?

Finally, an important missing ablation is a simple baseline using ALLO with cluster means as subgoals. Since the proposed method appears to extend ALLO with a more sophisticated subgoal planning strategy, this baseline would help isolate the contribution of the new approach.

**Questions:**

Novelty:
- Is the main novelty of your method that you have a new way to generate nodes for a planing algorithm based on a Laplacian Spectral Clustering Algorithm? How does this compare to other methods for deriving nodes from Laplacian Representation Learning algorithms. I believe using low-frequency eigencomponents is something that has been done, e.g. [1,2]
- Confirming that your use of a diffusion planner is not novel?

Experiments:
- I don't think I understand why you should expect generalization if you swap keyboard graphs across domains?
- How do I interpret Table 3? What is the baseline comparison? It's just your method, which is confusing without references.
- How do I interpret table 4? Same question for baseline comparisons? Should I just be comparing to your method when trained on the original target environment? There isn't a size dimension for Table 2 as far as I understand. The transfer here is really good. All above 95%. How do comparison methods do?
- For the visualization of keypoints, how do other methods for detecting keypoints work? This is relevant for Figure 3.
- A naive bottleneck discovery method would just use cluster means. I don't see that as comparison method. This seems important as the main contribution of this paper, as I understand it, is that you choose the "optimal" subgoal based on properties on properties of Laplacian spectral clustering.

[1] Proto-value Functions: A Laplacian Framework for Learning: Representation and Control in Markov Decision Processes
[2] A Laplacian Framework for Option Discovery in Reinforcement Learning

---

> ### Author Response · Authors · 2025-11-23
> **Response to Reviewer gT9f (Part 1/3)**
>
> ### W1 & Q1. It’s unclear what the specific contribution of this paper?
>
> We thank the reviewer for pointing out the close connection to Laplacian-based RL and for highlighting related work that uses Laplacian representations for goal-conditioned RL and subgoal discovery.
> First, we would like to clarify our intent. Our core contribution is not to propose a new Laplacian RL algorithm. Rather, as an offline goal-conditioned RL (OGCRL) paper, our main scientific contribution is to **rethink subgoal selection in the offline setting**. Most prior hierarchical or goal-conditioned methods generate subgoals via heuristics, e.g., fixed time windows or short-horizon reachability. Such subgoals typically lack semantic grounding: they do not explicitly mark the hard-to-cross bottlenecks in the state space that any successful trajectory to the goal must pass through, nor the key states at which the agent should switch behavior modes. As a consequence, agents often struggle exactly at these bottlenecks. For example, AntMaze agents colliding with corners and falling when turning, or Kitchen agents misaligning during grasps and object transport.
>
> In short, we point out that in long-horizon offline GCRL, **subgoals should not only plan short-term motion but should also explicitly anticipate how to traverse the next bottleneck and when to switch behavior modes**. Our work operationalizes this idea by using Laplacian representations as a tool: we apply Laplacian spectral clustering to offline data to expose metastable regions and their boundaries, treat the boundary crossings as keypoints, and construct a directed keypoint graph on top of them.
>
> We fully agree with the reviewer that Laplacian-based representations have already been shown to benefit RL, including for goal-conditioned settings and subgoal/option discovery,  we will revise relative wording. The distinction we aim to make is that most prior Laplacian-RL work leverages Laplacian structure primarily for online exploration or option discovery, whereas our focus is on better exploiting offline GCRL datasets through explicit bottleneck-based subgoal planning. Concretely, we (i) formalize that the next bottleneck is the optimal one-step subgoal under metastability assumptions, (ii) show that Laplacian spectral clustering recovers these bottlenecks with high overlap, and (iii) build a practical OGCRL algorithm that uses these bottleneck keypoints for routing purely from offline data, yielding substantial gains on D4RL and OGBench benchmarks.
>
> ---
>
> ### W2. How goals are defined?
>
> In the FrankaKitchen manipulation tasks, each evaluation instance specifies 4 object-centric goals (skills) chosen from a pool of 7 possible skills, and the particular 4-goal combinations used at test time do not appear in the offline dataset; the dataset only contains trajectories that achieve individual skills or partial combinations. For the OGBench navigation tasks (PointMaze / AntMaze / HumanoidMaze), each maze layout comes with 5 test tasks, and each task has a single goal. The offline dataset is collected with randomly sampled start and goal states that differ from those used at evaluation, so the policy is trained once on a single replay buffer and then evaluated on multiple unseen start–goal pairs.
> We would like to emphasize that our core contribution is a rethinking of subgoal selection for long-horizon offline GCRL, not a new benchmark setting: we use these standard environments exactly as defined in the original D4RL/OGBench configurations, and build our bottleneck-based hierarchical method on top of them.
>
> ---

---

> > ### Author Response · Authors · 2025-11-23
> > **Response to Reviewer gT9f (Part 2/3)**
> >
> > ### W3 & Q6, Q7. Comparison with naive cluster-based subgoals & visualization of the keypoints of other offline GCRL methods
> >
> > Motivated by the comment on ALLO + cluster means, we ran a small internal ablation under a shared ALLO representation on two representative long-horizon tasks. Concretely, we compared: (i) an ALLO-centroid baseline, which runs Laplacian spectral clustering and for each cluster picks a state near the cluster mean as a subgoal, and (ii) BASS (ours), which selects keypoints only near cluster boundaries (bottlenecks), as described in the paper.
> >
> > | Environment               | Centroid baseline (mean ± std, %) | BASS (ours, %, mean ± std) |
> > | ------------------------- | --------------------------------- | -------------------------- |
> > | antmaze-large-stitch-v0   | 40.7 ± 6.1                        | 81.0±7.0                   |
> > | pointmaze-large-stitch-v0 | 97.3 ± 1.2                        | 99.3 ± 1.2                 |
> >
> > Based on the table, on antmaze-large-stitch-v0, BASS improves success from 40.7±6.1% (centroid baseline) to 81.0±7.0%, nearly doubling performance. On the simpler pointmaze-large-stitch-v0, BASS still shows a slight edge, suggesting that bottleneck-aware subgoals help most when low-level dynamics make precise corridor traversal and turning challenging, while remaining mildly beneficial even in easier settings.
> >
> > Additionally, from the viewpoint of offline GCRL, a more direct and widely relevant comparison is to offline GCRL methods that also discover keypoints / landmarks directly from the raw states of an offline dataset, such as QPHIL ([https://arxiv.org/abs/2411.07760](https://arxiv.org/abs/2411.07760)). QPHIL trains a VQ-VAE on the raw state space and treats each codebook vector as a “landmark,” clustering states by Euclidean distance in the original coordinates and then planning over those discrete tokens. As shown in their own visualizations (e.g., on AntMaze), this leads to clusters that are largely geometric: states that are close in Euclidean distance are grouped together even when they are separated by walls and require long detours to reach each other in the true transition graph. In other words, the discovered “keypoints” do not necessarily align with actual bottlenecks or low-conductance boundaries.
> > In contrast, BASS leverages Laplacian spectral clustering, which produces keypoints with much clearer semantic meaning in terms of connectivity and bottlenecks. Fig. 3 shows that our keypoints concentrate in narrow corridors, doorways, and intersections, instead of being spread uniformly across regions or accidentally “bleeding” across walls.
> > To complement this visualization, we also compared numerically against a VQ-VAE–based method like QPHIL on standard D4RL AntMaze benchmarks:
> >
> > | Env                       | QPHIL  | BASS（ours）  |
> > | ------------------------- | ------ | ----------- |
> > | antmaze-medium-play-v2    | 91 ± 2 | 98.0 ± 0.0  |
> > | antmaze-medium-diverse-v2 | 92 ± 4 | 96.7 ± 0.9  |
> > | antmaze-large-play-v2     | 80 ± 3 | 96.0 ± 1.6  |
> > | antmaze-large-diverse-v2  | 82 ± 6 | 98.7 ± 1.9  |
> > | antmaze-ultra-play-v0     | 62 ± 4 | 97.3 ± 0.9 |
> > | antmaze-ultra-diverse-v0  | 62 ± 7 | 88.0 ± 1.6 |
> >
> > On these shared tasks, BASS consistently achieves higher success rates—for example, improvements of roughly +4–7 points on medium mazes, +16–17 points on large mazes, and 25+ points on ultra-scale mazes. This quantitative trend is consistent with the qualitative picture: naive geometric clustering (cluster means or VQ-VAE landmarks) tends to miss the true bottlenecks, whereas our Laplacian–bottleneck keypoints both (i) align with the intuitive bottleneck structure seen in Fig. 3 and (ii) translate into substantially stronger offline GCRL performance.
> >
> > ---

---

> > > ### Author Response · Authors · 2025-11-23
> > > **Response to Reviewer gT9f (Part 3/3)**
> > >
> > > ### Q3, 4, 5: Why should we expect generalization when swapping keypoint graphs across domains? How should Tables 3 and 4 be interpreted, and what is the baseline? How do comparison methods perform?
> > >
> > > Our goal with Tables 3 and 4 is not to introduce a new benchmark or to claim that BASS beats all prior methods in the transfer setting, but to diagnose what structure the learned keypoint graph actually captures.
> > >
> > > #### Part1. Why expect any generalization when swapping graphs?
> > >
> > > In all “graph swapping” experiments, we deliberately choose pairs of domains that share the same underlying 2D maze layout, but differ substantially either in data distribution or in dynamics:
> > > ** 1. AntMaze-Stitch vs AntMaze-Explore:**
> > > Same maze and navigation objective, but very different data:
> > > - Stitch contains high-return trajectory fragments stitched together;
> > > - Explore consists mostly of low-return, random-walk-like behavior.
> > >
> > > **2. PointMaze → AntMaze:**
> > > Same maze topology (rooms + corridors), but very different dynamics: a point mass versus a multi-legged ant.
> > >
> > > If the keypoint graph were just memorizing a particular dataset or behavior policy, we would expect performance to collapse once we (i) change the dataset (Stitch ↔ Explore) or (ii) change the dynamics (Point → Ant). Instead, we observe that:
> > > - Swapping graphs between Stitch and Explore preserves a large fraction of the success rate compared to using a graph learned on the target dataset itself.
> > > - Transferring a graph learned in PointMaze to AntMaze still yields high success, despite the much more complex ant dynamics.
> > >
> > > We interpret this as evidence that the graph is primarily capturing topological bottlenecks of the maze (rooms, corridors, junctions) that are invariant across datasets and to some extent across dynamics, rather than overfitting to a particular behavior distribution. We will clarify this diagnostic purpose more explicitly in §5.3 instead of presenting the results as a standalone “transfer benchmark”.
> > >
> > > ---
> > >
> > > #### Part 2. What is the baseline in Tables 3 and 4? How should they be read?
> > >
> > > In Table 3, we study high-level graph transfer. For each target environment, the baseline is BASS (in-domain), where both the keypoint graph and the low-level policy are trained on the same dataset. The other entries reuse the low-level policy of the target environment but swap in a keypoint graph learned from another dataset/domain (e.g., Stitch → Explore). The intended reading is simply: for each row, compare its success rate to the in-domain BASS row of the same environment; if performance remains close, this indicates that the transferred graph has captured reusable topological structure rather than overfitting to one dataset.
> > >
> > > In Table 4, we instead study low-level policy transfer. Here the keypoint graph is always learned in-domain for the target environment, and the baseline is again BASS (in-domain), where the low-level policy is also trained on that environment. The transfer rows keep the same keypoint graph but reuse a low-level controller trained elsewhere. The goal of these experiments is to show that once the high-level keypoint graph has accurately identified bottlenecks, the AntMaze low-level policy no longer needs to memorize the maze layout (walls, junctions, etc.), but can achieve high success by simply moving roughly straight between successive bottlenecks. We will clarify these baselines and labels in the revised tables and captions.
> > >
> > > ---
> > >
> > > #### Part 3. “The transfer here is really good (all >95%). How do comparison methods do?”
> > >
> > > For clarity, Tables 3 and 4 are not meant to compare BASS against other algorithms, but to compare different variants of BASS (in-domain vs. swapped high-level keygraphs/low-level policies). External baselines such as HIQL, Diffuser, etc., do not have a notion of an explicit keypoint graph that can be swapped between domains, so a direct “graph swap” comparison is not well-defined for them.
> > >
> > > Instead, the comparison to prior methods is done in Table 2, which reports standard in-domain performance (no graph transfer) on the same benchmarks.

---

### Official Review · Reviewer_PGk3 · 2025-11-01

**Soundness:** 3
**Presentation:** 2
**Contribution:** 2
**Rating:** 4
**Confidence:** 5

**Summary:**

* This paper proposes BASS (Bottleneck-Aware Spectral Subgoaling), an offline goal-conditioned reinforcement learning (OGCRL) framework that identifies bottleneck states which connect metastable regions in the state space and utilizes them as subgoals for hierarchical planning.

* BASS argues that the primary difficulty in long-horizon planning within OGCRL environments arises from hard-to-cross bottlenecks, rather than short-term value estimation. It formulates the subgoal selection problem as a spectral graph problem, leveraging the low-frequency structure of the Laplacian operator computed from offline data.

* BASS achieves superior performance compared to prior methods on D4RL and OGBench benchmarks. However, the method has only been validated on numeric-state environments, and the subgoal features appear to have been manually specified, which could further limit its applicability.

**Strengths:**

* The paper introduces a novel spectral perspective on hierarchical goal-conditioned reinforcement learning by framing subgoal discovery as a bottleneck identification problem in the state space.  BASS leverages the low-frequency eigenstructure of the Laplacian operator derived from offline data to reveal metastable regions and their connecting bottlenecks.

* The paper demonstrates theoretical analysis (Theorem 1 & 2) with a coherent algorithmic design that ties Laplacian representation learning, spectral clustering, and keypoint graph construction into a unified pipeline.

* By grounding subgoal discovery in topological structure rather than temporal heuristics, BASS offers a principled approach that could influence future research in hierarchical and representation-driven RL

**Weaknesses:**

1. The proposed method appears to be applicable only to numeric-state environments, and all experiments were conducted exclusively under such low-dimensional state settings. It remains unclear whether the approach can extend to high-dimensional visual-state environments (e.g., image-based observations), and whether the Laplacian-based embedding would still effectively identify bottlenecks in such cases.
Clarifying the applicability to more complex observation modalities would strengthen the generality of the contribution.

2. The paper represents each keypoint (KP) as (IΔ, vΔ), constraining only a subset of the state coordinates. However, the process for determining IΔ that is, which dimensions to include (e.g., x,y) appears to rely on manual or environment-specific heuristics. A more principled or learnable mechanism for feature selection (e.g., through information bottleneck criteria or gradient-based relevance analysis) would improve both generality and reproducibility.

3. In the GENERALIZATION ACROSS ENVIRONMENTS experiment, it is unclear what meaningful insight is gained by swapping keypoint graphs between AntMaze-Stitch and AntMaze-Explore. These two datasets share the same map and the same navigation objective, implying that their keypoint graphs should not differ substantially. Moreover, the reported cross-domain transfer between PointMaze and AntMaze is difficult to interpret: if subgoals are represented solely by (x,y) coordinates and the low-level controller also relies on these coordinates as its input, then the successful transfer is almost inevitable by design, rather than demonstrating genuine generalization.

4. This hand-crafted use of (x,y) as subgoal features may also undermine the fairness of comparison with prior baselines such as HIQL and Diffuser, which employ full-state embeddings (including joint angles or high-dimensional latent representations) for subgoal conditioning.

5. While the bottleneck-based subgoal concept is conceptually appealing, the paper lacks an in-depth comparison with temporal distance representation (TDR)-based approaches such as HILP, QRL, and GAS. TDR methods learn representations that reflect optimal time distances—allowing farther movement in open regions and shorter transitions in constrained areas like corners. Since these works also utilize temporally grounded structure for subgoal selection or graph construction, a quantitative comparison would help clarify the specific advantages of BASS’s spectral formulation.

6. Finally, although BASS demonstrates improvements on numeric-state benchmarks, the paper does not analyze failure cases.
It remains ambiguous whether the failures arise from (i) imperfect keypoint extraction or graph connectivity, (ii) limited capability of the low-level controller, or (iii) excessive distances between consecutive keypoints. For instance, in AntMaze-giant-stitch and AntMaze-large-explore, the success rates drop significantly. It would be insightful to analyze these cases in light of concurrent work such as GAS (Baek et al., 2025), which achieves strong performance using a TDR-based graph representation. Such analysis could reveal whether BASS’s limitations stem from the keypoint discovery mechanism itself or from broader challenges in long-horizon offline control.

**Questions:**

Please provide responses to the weaknesses mentioned above.

---

> ### Author Response · Authors · 2025-11-23
> **Response to Reviewer PGk3 (Part 1/4)**
>
> ### Q1. Applicability to visual / high-dimensional observations
>
> Thank you for raising this important point. In the main paper we focused on numeric-state benchmarks, which can indeed give the impression that BASS is restricted to low-dimensional inputs. We would like to clarify that BASS is not limited to numeric states and can be applied to high-dimensional visual observations as well.
>
> Concretely, we additionally evaluated BASS on the visual-antmaze-large-navigate and visual-antmaze-large-stitch environments, where the agent observes images. In these experiments, we follow the standard setup: images are first encoded into vector observations via a visual encoder, and we then build the Laplacian graph and perform bottleneck/keypoint discovery in this latent state space, using exactly the same BASS pipeline.
>
> The results (success rate, %, averaged over 3 seeds) are:
>
> | Environment                   | BASS (ours) | HIQL       |
> | ----------------------------- | ----------- | ---------- |
> | visual-antmaze-large-navigate | 78.7 ± 2.3  | 53 ± 9 |
> | visual-antmaze-large-stitch   | 68.0 ± 2.0  | 28 ± 2 |
>
> Even in these high-dimensional visual settings, BASS significantly outperforms the HIQL baseline. This suggests that Laplacian-based bottleneck identification remains effective when applied to latent representations of images, and that BASS does not rely on the state space being low dimensional.
>
> Intuitively, BASS operates on the reachability topology induced by trajectories (rooms, corridors, bottlenecks, etc.), rather than on the raw coordinate system of the observations. As long as the visual encoder preserves the temporal/transition structure in its latent space, our Laplacian spectral clustering and bottleneck keypoint extraction can be applied in exactly the same way.
>
> ---
>
> ### Q2. How is $(I_\Delta)$ (the subset of constrained coordinates) chosen? Is it hand-crafted or environment-specific?
>
> Thank you for this thoughtful comment. We would like to clarify that in our implementation $(I_\Delta)$ is not hand-crafted, but determined by a simple, fully automatic procedure inspired by Laplacian RL.
>
> Intuitively, the Laplacian representation is designed to capture low-frequency, slowly varying structure of the dynamics. We therefore restrict the Laplacian construction to state coordinates that themselves evolve slowly and smoothly along trajectories, and automatically drop coordinates that fluctuate quickly and irregularly (e.g., high-frequency joint angles). Concretely:
>
> * For each state dimension, we measure its temporal smoothness over the offline dataset (e.g., how large the typical change $(s_{t+1}^i - s_t^i)$ is along trajectories, and how continuous its evolution is).
> * Dimensions with low temporal variation / strong continuity are kept in $(I_\Delta)$; dimensions with high-frequency, noisy changes are excluded from the Laplacian embedding.
> * This procedure is entirely automatic across environments, and does not rely on environment-specific hand-engineered rules.
>
> Finally, we note that BASS continues to outperform HIQL in visual AntMaze environments where the input is the full visual embedding (see our answer to Q1): this suggests that the core benefit of BASS comes from bottleneck-based subgoal discovery, not from delicate manual design of which coordinates to constrain.
>
> ---

---

> > ### Author Response · Authors · 2025-11-23
> > **Response to Reviewer PGk3 (Part 2/4)**
> >
> > ### Q3. What is the insight from swapping keypoint graphs across environments? Is PointMaze → AntMaze transfer trivial since both use (x,y)?
> >
> > Thank you for these questions. The goal of these experiments is not to claim a new benchmark result, but to probe what kind of structure the keypoint graph learned by BASS is actually capturing. We agree that this motivation is not sufficiently clear in the current draft and will clarify it in the revision.
> >
> > #### (a) AntMaze-Stitch ↔ AntMaze-Explore: same map, different data
> >
> > Although AntMaze-Stitch and AntMaze-Explore share the same underlying maze layout and navigation objective, their offline datasets are drastically different:
> >
> > * AntMaze-Stitch contains many high-quality fragments from a reasonably controller.
> > * AntMaze-Explore is dominated by low-return, random-walk–like behavior with poor coverage near optimal paths.
> >
> > In Table 3, we deliberately decouple the keypoint graph from the low-level controller:
> >
> > * For each target dataset, the low-level goal-conditioned policy is always trained on that dataset itself.
> > * We then only swap the keypoint graph: e.g., use a graph learned from Stitch to provide bottleneck-based subgoals for the low-level policy trained on Explore.
> >
> > If the keypoint graph were merely overfitting to behavior-specific statistics or reward shaping (e.g., “where the good trajectories happen to pass”), then transplanting a graph from a qualitatively different dataset should severely hurt performance. Instead, we observe that:
> >
> > * Swapping graphs between Stitch and Explore yields only moderate degradation compared to using a graph learned on the same dataset; performance does not collapse.
> >
> > We interpret this as evidence that the graph is primarily encoding topological bottlenecks of the maze (rooms, corridors, junctions) that are invariant across behavior distributions, rather than encoding idiosyncrasies of a particular policy.
> >
> > #### (b) PointMaze → AntMaze transfer: why it is not trivial
> >
> >
> > 1. What is transferred?
> >
> >    * We transfer the keypoint graph learned in PointMaze, i.e., the set of bottleneck subgoals and their connectivity on the 2D layout.
> >    * The low-level controller in AntMaze is trained from scratch on AntMaze data.
> >
> > 2. Why this is non-trivial:
> >
> >    * PointMaze uses simple point-mass dynamics, while AntMaze uses a high-dimensional quadruped (ant) dynamics; the low-level controller still has to learn to navigate with very different physics.
> >    * The transferred keypoint graph only provides a sparse sequence of bottleneck waypoints. If the graph learned in PointMaze did not align with the true corridors and chokepoints that are feasible under ant dynamics, the low-level policy would not be able to consistently reach the final goal, and success rates would drop sharply.
> >    * In practice we observe that using a PointMaze-derived keypoint graph in AntMaze achieves success rates close to the in-domain BASS baseline, which indicates that the graph is capturing map-level bottlenecks that remain useful even when the underlying dynamics change.
> >
> > We also observed empirically that keypoint graphs learned in PointMaze tend to be slightly “cleaner” than those learned directly in AntMaze: point-mass dynamics are smoother and more stable, which leads to clearer bottleneck structure in the Laplacian representation. This suggests an interesting future direction: transferring bottleneck structure learned in simpler systems to more complex ones. For example, in robotic manipulation, one could first learn a bottleneck/keypoint graph with a simpler two-finger gripper in an easy environment, and then reuse that graph as a topological prior when training a more complex five-finger hand.
> >
> > ---
> > ### Q4. Fairness of using (x,y) as subgoal features vs. full-state conditioning in HIQL / Diffuser
> >
> > We would like to clarify that the subgoals we provide to the low-level controller are not restricted to (x, y) coordinates, but are full states, which keeps the comparison with baselines such as HIQL and Diffuser fair.
> >
> > Concretely, the high-level module first performs Laplacian spectral clustering, then identifies boundary states between clusters and computes the mean state over boundary states for each cluster interface; this mean state is treated as a keypoint. Each such keypoint is a full state vector: it includes not only the position (x, y), but also all joint angles and any other high-dimensional latent components. This entire state is then used as the subgoal fed into the low-level policy.
> >
> > This is exactly analogous to HIQL and Diffuser, which also condition their policies on full-state (or full-latent) subgoals rather than on (x, y) alone. The only place where (x, y) plays a special role is in the representation used for constructing the Laplacian, where we use a simple, fully automatic rule to focus on slowly varying coordinates, but the actual subgoal passed to the controller always contains the complete state information.
> >
> > ---

---

> > > ### Author Response · Authors · 2025-11-23
> > > **Response to Reviewer PGk3 (Part 3/4)**
> > >
> > > ### Q5 On the relation to TDR-based methods (QRL, HILP, GAS) & Q6 failed case analysis
> > >
> > > We agree that TDR-based methods are highly relevant. While QRL, HILP, and GAS all learn some notion of state–state reachability (typically the expected number of steps under an optimal policy), this signal is mainly used to support uniform, fine-grained planning, i.e., generating dense waypoint sequences along a trajectory, rather than to explicitly isolate the small set of critical bottleneck states that determine whether long-horizon goals are reachable. This lack of explicit focus on bottlenecks can make these methods vulnerable precisely at narrow passages; for example, in antmaze-giant-navigate, GAS achieves only around 40% success on one particular task, despite an overall score of 77.6%, because the planned waypoint path passes extremely close to wall corners at several junctions of this task, leading to frequent collisions and falls. BASS is designed exactly around these sparse bottlenecks and their role in long-horizon planning.
> > >
> > > **Part1 Comparison with QRL and HILP**
> > >
> > > Concretely, QRL learns a quasi-metric / temporal-distance function between states and uses it to regularize value learning and planning, but it does not build a discrete bottleneck graph or identify semantic subgoals where behavior should switch. HILP learns a temporal latent model and plans in that latent space; subgoals are chosen as evenly spaced latent states (or segments) along a trajectory, and the experiments focus mainly on relatively simple 2D navigation (e.g., Maze2D). In our work, QRL is already included as a baseline in Table 2, and BASS consistently outperforms it on shared benchmarks. For HILP, on the common Maze2D-large-play setting our performance is comparable to the reported HILP results, while our method additionally scales to substantially more challenging navigation and manipulation tasks (AntMaze, HumanoidMaze, and FrankaKitchen).
> > >
> > > **Part 2. Diagnosing and improving weaker environments**
> > >
> > > For the two weaker environments highlighted by the reviewer, we performed a more detailed analysis and found that the limitations stem from concrete, fixable design choices rather than a fundamental flaw in the keypoint mechanism.
> > >
> > > 1. **AntMaze-giant-stitch.**
> > >    In our original submission, the clustering number $(K)$ for AntMaze-giant-stitch was mistakenly set much lower than for other giant mazes, which led to incomplete coverage of bottlenecks: some critical corridor junctions were never assigned any keypoint. As a result, the low-level controller was forced to traverse “unmarked” bottlenecks without guidance, causing failures. After correcting this and setting $(K)$ to match the value used in pointmaze-giant (i.e., $(K=50)$), the performance on antmaze-giant-stitch improves from $(20)$ to **(71.3 ± 7.0)**. This supports the interpretation that the original failure was due to under-covering bottlenecks, not to an inherent limitation of BASS.
> > >
> > > 2. **AntMaze-large-explore.**
> > >    For antmaze-large-explore, the main bottleneck lies in the **low-level controller** rather than in the high-level planning. Specifically, we use a HIQL-like MLP “keypoint regressor” that predicts an intermediate state $(\tilde{s}_{t+k})$ to stabilize and shorten the planning horizon. In the initial experiments, we set the planning horizon $(k)$ (way_step) to a very small value (5). Under the low-quality, noisy explore data, this short horizon does not provide a strong enough training signal for the MLP to learn robust motion primitives, leading to poor execution between keypoints. When we increase $(k)$ from 5 to 25, the performance on antmaze-large-explore rises from $(42)$ to **(72.7 ± 1.2)**, indicating that the high-level bottleneck graph is still useful, and the original weakness is mainly due to an underpowered low-level configuration on noisy data.
> > >
> > > These diagnostics suggest that the observed failures correspond to (i) insufficient bottleneck coverage when $(K)$ is set too low (AntMaze-giant-stitch) and (ii) inadequate low-level horizon on noisy datasets (AntMaze-large-explore), rather than a structural limitation of bottleneck-based subgoal discovery. We will report these updated results in the revised version.

---

> ### Author Response · Authors · 2025-11-23
> **Response to Reviewer PGk3 (Part 4/4)**
>
> **Q5 & Q6 - Part3 Relation to GAS.**
>
> Among TDR-based methods, GAS is conceptually closest, also using reachability graphs for planning. However, their objectives differ sharply: GAS builds dense graphs with thousands of nodes at fixed temporal-distance intervals—most being ordinary waypoints lacking decision semantics. In contrast, BASS discovers a sparse set of dynamical bottlenecks (≈10–50 keypoints, i.e., "rooms × doors"), each representing an interpretable decision point where behavior modes change (entering/exiting corridors, passing junctions). Thus, GAS optimizes micro-level planning while BASS provides macro-level guidance.
>
> Empirically, on shared benchmarks our method achieves competitive performance with GAS in complex manipulation tasks such as kitchen-partial, and on navigation tasks our scores are close to the reported GAS results when its graph is constructed without aggressive TDR-based data filtering. We attribute the remaining performance gap primarily to two design choices: (i) GAS uses a very dense TDR graph and shortest-path planning, which substantially lowers the difficulty of navigation tasks, whereas BASS deliberately keeps only a few dozen sparse KPs and leaves local motion between bottlenecks to the low-level policy; and (ii) GAS heavily optimizes the low-level layer via reward shaping and data cleaning (retaining only 2–8% of high-quality transitions), while BASS intentionally uses simple, plug-and-play low-level controllers (e.g., vanilla MLP / diffusion planners) without any data filtering or extra intrinsic rewards, This is because the core contribution of BASS is to isolate and study the value of bottleneck-based subgoal selection itself, rather than to engineer the strongest possible low-level controller via data curation and reward shaping.
>
> Thus, we view BASS and GAS as **complementary**: GAS enables fine-grained waypoint planning and low-level training, while BASS identifies macro subgoals determining where agents must switch modes. Indeed, GAS failures often stem from not respecting bottlenecks (e.g., waypoints hugging corners). A natural future direction is to combine both: use BASS bottlenecks as mandatory macro waypoints, and let GAS's TDR graph refine micro-level paths within regions. This fusion would enrich the offline GCRL toolbox and unlock new algorithmic designs, highlighting the potential of reachability representations across hierarchy levels.

---

> ### Comment · Reviewer_PGk3 · 2025-11-27
>
> I will revise my score, as the authors’ responses and additional experimental results have sufficiently addressed the concerns and weaknesses I previously identified.

---

> > ### Author Response · Authors · 2025-11-27
> > **Thank you for your updated evaluation**
> >
> > Thank you very much for your thoughtful review and for taking the time to re-evaluate our paper after reading our rebuttal. We are very glad that our responses helped address your concerns, and we truly appreciate your positive assessment of our work.

---

### Author Response · Authors · 2025-12-03
**Summary of Rebuttal Updates**

We would like to express our sincere gratitude to the Area Chair and all four reviewers for their careful assessment of our work. Below we provide a brief summary of the rebuttal process and its impact, for your reference.

The initial scores for our submission were 4-6-4-4. During the rebuttal phase, **two reviewers (PGk3 and 3mmM) engaged in the discussion and both raised their initial scores from 4 to 6**, explicitly stating that our responses “sufficiently addressed the concerns and weaknesses.” Thus, before the rebuttal phase was frozen, the overall scores had improved **from 4-6-4-4 to 6-6-4-6**. Our follow-up thank-you messages to these two reviewers also corroborate this development.

All rebuttal-related comments and responses, including the subsequent score increases, were completed before the OpenReview information leakage incident occurred. Both the reviews and our author responses carry explicit timestamps, which clearly show that all discussions and score updates took place prior to the incident. Neither our rebuttal text nor any updated reviewer comments relied on, or referred to, any external or unauthorized information.

---

For the two reviewers who **did not reply** further during the rebuttal, we carefully revised the paper in light of all concerns raised in their initial reviews, as summarized below.

---

For the non-responding 4-score reviewer Ne7X, their initial review already clearly acknowledged the novelty, experimental results, and visualizations of our method. Their suggestions were mainly incremental improvements, including:

(1) extending the ablation study over the number of clusters

(2) adding comparisons against the baselines in terms of trajectory steps

(3) discussing one additional related work

(4) providing visual comparisons with oracle trajectories

(5) adding pseudocode for key components of the method.

During the rebuttal, we systematically added extensive experiments and revised the paper around these points, addressing each suggestion one by one. Notably, the highest-priority and most central issue (1) the extended ablation experiments and discussion on the number of clusters, **was also raised by reviewer 3mmM and has already been explicitly acknowledged in their updated review, accompanied by an increased score**.
Although, due to changes in the review policy, Ne7X did not have the opportunity to post further comments or continue the discussion, given their overall positive initial evaluation and the improvements we have implemented, **we believe their main concerns have been substantively addressed and that they might reasonably have considered raising their score**.

---

For the non-responding 6-score reviewer gT9f, we similarly provided comprehensive clarifications and additions, including:

(1) building on their positive view of our contribution (which is also shared by the other three reviewers) by further elaborating the technical innovations of our method as an offline goal-conditioned RL approach

(2) introducing additional comparative experiments with other clustering methods (such as QPHIL)

(3) clarifying the setup and details of our generalization experiments.

Importantly, **points (2) and (3) were also raised by reviewers 3mmM and PGk3, respectively, and our corresponding rebuttal and revisions were explicitly acknowledged by them, again accompanied by score increases**.

 **Taken together, these changes substantially address the issues raised in gT9f’s review and, had further discussion been possible, might reasonably have led to a more positive overall assessment on their side**.

---

In summary, the rebuttal process has already led to clear score improvements from two reviewers and, For the remaining 4-score reviewer, who did not have time to engage further before the review policy change, we believe we have also **addressed all of their concerns**.

We once again thank you for your efforts as Area Chair and would be grateful if you could take the above rebuttal process and its positive outcomes into consideration when making the final decision.

---

### Note · Program_Chairs · 2026-01-17
**Submission Desk Rejected by Program Chairs**

The following references in this submission do not refer to real documents and/or have major errors in bibliographic information:

 Zeyu Zeng, Yifei Zeng, Ruida Liu, et al. Goal-conditioned predictive coding for offline reinforcement learning. In Advances in Neural Information Processing Systems (NeurIPS),
2023. URL https://proceedings.neurips.cc/paper_files/paper/2023/file/51053d7b8473df7d5a2165b2a8ee9629-Paper-Conference.pdf.